# Scalable 3D Object-centric Learning

## Abstract

We tackle the task of unsupervised 3D object-centric representation learning on scenes of potentially unbounded scale. Existing approaches to object-centric representation learning exhibit significant limitations in achieving scalable inference due to their dependencies on a fixed global coordinate system. In contrast, we propose to learn view-invariant 3D object representations in localized *object coordinate systems*. To this end, we estimate the object pose and appearance representation separately and explicitly project object representations across views. We adopt amortized variational inference to process sequential input and update object representations online. To scale up our model to scenes with an arbitrary number of objects, we further introduce a *Cognitive Map* that allows the registration and querying of objects on a global map. We employ the object-centric neural radiance field (NeRF) as our 3D scene representation, which is jointly inferred by our unsupervised object-centric learning framework. Experimental results demonstrate that our method can infer and maintain object-centric representations of unbounded 3D scenes. Further combined with a per-object NeRF finetuning process, our model can achieve scalable high-quality object-aware scene reconstruction.

## 1 Introduction

The ability to understand 3D surroundings in an object-centric way is crucial for AI agents to perform a range of tasks including relational reasoning (Chang et al., 2017) and reinforcement learning (Diuk et al., 2008). In recent years, 2D and 3D unsupervised object-centric learning have attracted increasing attention in the field. Compared with 2D object-centric learning methods (Eslami et al., 2016; Lin et al., 2020; Burgess et al., 2019; Crawford & Pineau, 2019; Locatello et al., 2020) that focus on decomposing 2D images into objects, 3D learning methods aim to recover the complete 3D scene structures in an object-centric way using RGB or RGBD observations (Li et al., 2020; Stelzner et al., 2021; Chen et al., 2021; Henderson & Lampert, 2020). A key limitation of existing 3D methods is that they can only handle scenes with a scale that can fit into the field of view (FOV) of a fixed number of cameras (Li et al., 2020; Stelzner et al., 2021; Chen et al., 2021), where the camera coordinates are specified within a global coordinate system. The trained inference models are thus highly dependent on the chosen global coordinate system (Li et al., 2020; Chen et al., 2021; Kabra et al., 2021; Eslami et al., 2018; Henderson & Lampert, 2020), and cannot generalize beyond the scale of the training sets. All of these limits the applicability of existing 3D methods to real-world problems or even simulated reinforcement learning environments (Beattie et al., 2016), where scenes with large or even unbounded scales are routinely encountered.

In this paper, we propose *Scalable Online Object Centric network in* **3D** *(SOOC3D)*. SOOC3D addresses the problem of scalability by inferring object poses and view-invariant object representations in localized *object coordinate systems* from RGBD data. To handle sequential data for large-scale scenes, we exploit amortized variational inference for online inference. Inferred object poses allow object representations to be explicitly projected across views with preserved identities. To keep track of all the detected objects throughout the online update, we introduce a highly scalable external memory mechanism named *Cognitive Map*,[1] which can be used to dynamically register and query detected object representations. This memory mechanism further removes a constraint in existing works (Henderson & Lampert, 2020; Burgess et al., 2019; Locatello et al., 2020; Engelcke et al., 2020a; Yu et al., 2022) whereby the maximum number of objects allowed in each scene is capped.

---

[1]The term cognitive map is borrowed from cognitive psychology studies on mental representations of the spatial surroundings in animal, and human brain (Kitchin, 1994).

We adopt the 3D object-aware Neural Radiance field (NeRF) to decode such representations to 3D geometries for training. While per-scene NeRF with direct SGD optimization can capture detailed 3D scenes (Mildenhall et al., 2021; Zhang et al., 2022; Tancik et al., 2022), the reconstruction quality of such unsupervised object-centric NeRF learning methods commonly falls short of the per-scene NeRF approaches as the introduced information bottlenecks filter out high-frequency information (Engelcke et al., 2020b). To narrow the gap, we introduce the per-object NeRF finetuning process to improve the reconstruction quality while preserving the objects' identities.

Our contributions are summarised as follows. i) We propose, to the best of our knowledge, the first unbounded scalable generative-model-based unsupervised 3D object-centric learning framework. ii) We learn the explicit object poses and view-invariant object representations separately via the amortized variational inference framework to achieve scalable online updating. iii) To store a potentially unbounded number of objects detected for scalable inference, we introduce *Cognitive Map* separating object representations management from the inference process. iv) We demonstrate that the reconstruction quality can be further improved via our per-object NeRF finetuning process with preserved objects' identities.

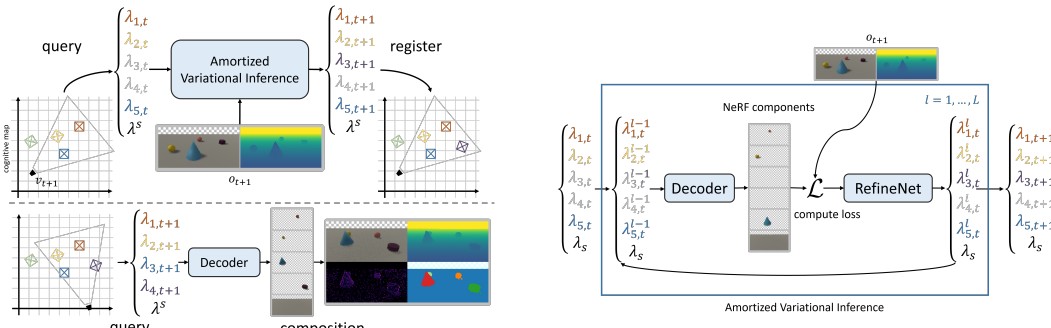

Figure 1: Left: The interaction between the inference pipeline and the cognitive map for scene updating (top) and novel view synthesis (bottom). Previously detected objects are registered in the cognitive map. When a new view arrives, the representations of all objects present in the view are retrieved. $\{\lambda_i\}$ are the object representations and $\lambda^s$ is the scene layout representation. If the number of objects existing in the current view is less than a pre-defined value $K$, we add prior representations (greyed $\lambda_i$). We update $\{\lambda_i\}$ to integrate new information with the amortized variational inference process and register them back into the cognitive map. For novel view synthesis, the retrieved representations are decoded into NeRF components. By components composition, we synthesize RGB image, depth, segmentation and uncertainty map. Right: A $L$-iteration amortized variational inference process. In each iteration, the set of input representations is decoded into NeRF components. We evaluate the likelihood of observation under the composed NeRF. The refinement network takes the decoded NeRF, observation and likelihood to update the object-centric representation.

## 2  RELATED WORK

**Unsupervised Generative Model-based 2D Object-centric Learning.** 2D object-centric learning aims to group pixels covering the same object under the same label and at the same time produce a neural representation of each discovered object. At the core of those methods is the spatial mixture model formulation that frames object-centric learning as a latent variable inference problem (Eslami et al., 2016; Greff et al., 2019; 2017). To handle observations with a high object density, a branch of works (Eslami et al., 2016; Crawford & Pineau, 2019; Lin et al., 2020) infers latent variables for local regions of each 2D observation. Pipelines equipped with iterative refinement modules (Greff et al., 2017; Locatello et al., 2020) refine the latent variable iteratively conditioned on an input view. Particularly, IODINE (Greff et al., 2019) employs amortized variational inference (Marino et al., 2018) that can process sequential data. However, the aforementioned methods do not infer 3D structures. Object latents are discarded once out of view.

**Unsupervised Generative Model-based 3D Object-centric Learning.** 3D-aware methods not only try to factorize observations in an object-centric manner but also infer the 3D spatial structure of scenes, which can be examined by the means of novel views synthesis. Similar to its 2D counterpart,

3D object-centric representation learning approaches also adopt the spatial mixture model formulation. ObSuRF (Stelzner et al., 2021) and uORF (Yu et al., 2022) introduce neural radiance fields (NeRF) into the object-centric learning setting. Smith et al. (2022) propose to use object light field to avoid dense sampling along rays during rendering. As an attempt to model scenes on a larger scale, O3V (Henderson & Lampert, 2020) and SIMONe (Kabra et al., 2021) infer object-centric representation from a video sequence. However, both O3V and SIMONe adopt a non-incremental method and process entire video sequences before generating scene representations. MulMON (Li et al., 2020) adopts amortized variational inference framework for 3D object-centric learning that allows object latents to be updated by new views in an online fashion.

The methods mentioned above work well on scenes with bounded sizes that can fit into the FOV of camera cones. One limiting factor of scaling up to larger scenes is that they take camera poses and locations specified in a pre-defined global coordinate system as input to the network. Thus, during test time, they cannot handle scenes larger than the ones in training sets. Our work infers view-invariant object representations that can scale up to unboundedly large scenes.

**Object-Compositional and Scalable NeRF.** Object-compositional NeRF has been studied recently to learn the 3D representation for each object and the scene for image synthesis. In particular, Yang et al. (2021) introduced a two-pathway framework to model the foreground objects and the scene branch, with known coarse object instance masks. Such a method cannot be directly applied to the unsupervised scenario. To handle large scenes, block-wise NeRF has been proposed recently(Zhang et al., 2022; Tancik et al., 2022). They can either generalize to large scene (Zhang et al., 2022) by taking multiple images as input or train scene-specific block-wise NeRFs for large scene fast rendering (Zhang et al., 2022). However, both Zhang et al. (2022) and Tancik et al. (2022) are not object-centric. In this paper, we aim to combine merits from all sides and learn scalable object-centric NeRF scene representations.

## 3 METHOD

With the assumption that the scene is static and there can be at most $K$ objects in each view (but unlimited in the entire scene), SOOC3D aims to achieve object-centric 3D scene understanding in unbounded scale during test time from sequential RGBD inputs. This goal requires the inference process to be online and view-invariant. In SOOC3D, the online inference is facilitated by the recurrent variational proposal distribution (Sec. 3.1). SOOC3D ensures a view-invariant object representation by decoding NeRF under the object coordinate system. The refinement network takes both view-dependent observation and view-invariant object representation to update the object latent variables (Sec. 3.2). Notably, the set of objects in view may change when we switch from one camera to another. To keep track of all object representations and only expose those that are in view to the inference process, we introduce a *Cognitive Map* for scalable learning (Sec. 3.2). The illustration of the inference pipeline is shown in Fig. 1. Finally, we show that object reconstruction quality can be improved with preserved identities through the per-object NeRF finetuning process with an object-centric initialization (Sec. 3.2).

### 3.1 GENERATIVE MODEL FORMULATION AND THE OPTIMIZATION OBJECTIVE

**Generative Model.** At any time $t$, we have access to a set of camera views $\mathcal{V}_t = \{v_1, \ldots, v_t\}$ specified by their extrinsic parameters. Each camera captures a RGB $c_{v_t}$ observation, and a depth observation $d_{v_t}$. We define $o_{v_t} = \{c_{v_t}, d_{v_t}\}$ and $\mathcal{O}_t = \{o_{v_1}, \ldots, o_{v_t}\}$. Below we hide the subscript of camera pose in equations that hold for all camera poses. We assume there are at most $K$ objects in one view. The $K$ object latent representations form a set $\mathcal{Z}_v^{obj} = \{z_{\phi_v(1)}^{obj}, \ldots, z_{\phi_v(K)}^{obj}\}$, where $\phi_v(\cdot)$ returns the global index of each object latent given its in-view index. The set of all object latent variables at time $t$ is $\mathcal{Z}_t^{obj} = \{z_1^{obj}, \ldots z_N^{obj}\} = \bigcup_{v \in \mathcal{V}_t} \mathcal{Z}_v^{obj}$.

The latent variable of an object is specified as $z_i^{obj} = \{z_i^{where}, z_i^{what}, z_i^{pres}\}$, where $z_i^{where} \in \mathbb{R}^3$ encodes object location on xz-plane and rotation around y-axis, $z_i^{what} \in \mathbb{R}^D$ encodes object appearance. $z_i^{pres} \in \{0, 1\}$ is a binary variable with $z_i^{pres} = 0$ indicating that object $i$ does not exist and $z_i^{pres} = 1$ otherwise. We further define $z_v^{scene} \in \mathbb{R}^D$ as the latent variable to model the scene layout under camera $v$. We assume that for each view, the scene layout component always exists and

the location of a scene layout component is always at the center of the camera view. We denote $\mathcal{Z}_v$ as $\mathcal{Z}_v^{obj} \bigcup z_v^{scene}$. The complete data likelihood function is defined as:

$$\mathcal{L} = \mathbb{E}_{v, \mathcal{Z}_v} \left[ p(o_v | \mathcal{Z}_v) \right] \tag{1}$$

**Amortized Variational Inference.** The exact object latent posterior $p(z_{i,t}^{obj} | \mathcal{O}_t, \mathcal{V}_t)$ is intractable. Thus, we resort to the amortized variational inference (Greff et al., 2019; Li et al., 2020; Emami et al., 2021). We approximate the true posteriors by a proposal distribution

$$q(z_{i,t}^{obj} | \mathcal{O}_t, \mathcal{V}_t) = \int_{z_{i,t-1}^{obj}} q(z_{i,t}^{obj} | o_{v_t}, v_t, z_{i,t-1}^{obj}) q(z_{i,t-1}^{obj} | \mathcal{O}_{t-1}, \mathcal{V}_{t-1}) dz_{i,t-1}^{obj}, \tag{2}$$

with $q(z_{i,0}^{obj} | \mathcal{O}_0, \mathcal{V}_0)$ being the variational prior. The recursive nature of the variational posterior enables online inference with a constant memory footprint. When object $i$ is not visible in view $v_t$, we define $q(z_{i,t}^{obj} | o_{v_t}, v_t, z_{i,t-1}^{obj})$ to be the Dirac delta function $\delta_{z_{i,t-1}^{obj}}(z_{i,t}^{obj})$. The posterior will remain unchanged as $q(z_{i,t}^{obj} | \mathcal{O}_t, \mathcal{V}_t) = q(z_{i,t-1}^{obj} | \mathcal{O}_{t-1}, \mathcal{V}_{t-1})$.

As an approximation, we further assume that $z_i^{where}, z_i^{what}, z_i^{pres}$ are independent of each other conditioned on $\mathcal{O}_t, \mathcal{V}_t$. We parameterize $q(z_i^{where})$ and $q(z_i^{what})$ as isotropic Gaussian with $\lambda_i^{where} = \{\mu_i^{where}, \sigma_i^{where}\}$ and $\lambda_i^{what} = \{\mu_i^{what}, \sigma_i^{what}\}$. $q(z_i^{pres})$ takes the form of Bernoulli distribution and the $\lambda_i^{pres}$ is the logit. Following the amortized variational inference, for each input view $v_t$, we update our latent for $L$ iterations. At the $l \in \{0, 1, \cdots, L\}$ iteration,

$$z_{i,t}^{obj,l} \sim q_{\lambda_{i,t}^l}(z_{i,t}^{obj,l} | o_{v_t}, v_t, z_{i,t}^{obj,l-1}) \tag{3}$$

$$\lambda_{i,t}^l = \lambda_{i,t}^{l-1} + f_\vartheta(z_{i,t}^{obj,l-1}, o_{v_t}, v_t, \mathbf{a}) \tag{4}$$

with $q_{\lambda_{i,t}^0} = q_{\lambda_{i,t-1}^L}$. $f_\vartheta$ is the refinement network and $\mathbf{a}$ is a collection of auxiliary input. For each view, we compute the KL-Divergence $\mathcal{L}_{v_t}^{kl} = \sum_{l=1}^L \mathcal{D}_{KL}[q_{\lambda_{i,t}^l} || q_{\lambda_{i,t}^0}]$. The iteration is executed in parallel for all object latent distributions that are detected in the current view.

During training, for each scene, in addition to $T$ input views $\mathcal{V}_T$, we also sample a set of query views $\mathcal{Q}$. The final Evidence Lower Bound (ELBO) is defined as

$$\mathcal{L} = \frac{1}{T} \sum_{t=1}^T \mathbb{E}_{q_{\lambda_t}} \left[ \log p(o_{v_t} | \mathcal{Z}_{v_t}) \right] + \frac{1}{|\mathcal{Q}|} \sum_{v \in \mathcal{Q}} \mathbb{E}_{q_{\lambda_T}} \left[ \log p(o_v | \mathcal{Z}_v) \right] - \frac{1}{T} \sum_{t=1}^T \mathcal{L}_{v_t}^{kl} \tag{5}$$

We adopt the same pre-ray likelihood function as Stelzner et al. (2021) to compute $p(o_v | \mathcal{Z}_v)$. For the completeness of the paper, we detail the likelihood function computation in Appendix B.

### 3.2 MODEL IMPLEMENTATION

**Coordinate System Transformation.** The view-invariant inference depends heavily on the coordinate system transformation between **camera coordinate systems** and the **object coordinate systems**. For each step, our model expects an RGBD observation and latent variables $\mathcal{Z}$ from the previous step. As detailed in Appendix B, along each ray $r$, we get the surface sample at the observed depth $d$, and one air sample with a depth less than $d$. We specify each sample $x$ in the camera coordinate system.

Each $z_{\phi_v(k)}^{where}$ is interpreted as an object pose in the current camera coordinate system. With the object poses specified, for each object indexed by $k$, we can build a projection matrix $\Pi(z_{\phi_v(k)}^{where}) \in SE(3)$ and map each point $x$ to the object local coordinate system by $x_k = \Pi(z_{\phi_v(k)}^{where}) \cdot x$ for decoding. The global coordinate system is only used for object registration and query.

**NeRF decoding.** Following common practice, we transform the coordinates of samples into harmonic representations (Mildenhall et al., 2021). Conditioned on $z_{\phi_v(k)}^{what}$ or $z_v^{scene}$, a NeRF decoder $\tilde{\sigma}_\theta(\cdot)$ assigns each point a raw density $\tilde{\sigma}_k(x_k) = \tilde{\sigma}_\theta(x_k, z_{\phi_v(k)}^{what}) \in [0, 1]$ and a RGB color. While the $K$ object components share the same NeRF decoder, the scene layout is decoded via its own decoder.

To accurately predict object location, we introduce an inductive bias in the form of Gaussian weighting. To be more precise, we compute the weighted density $\log \hat{\sigma}_k(x_k) = \log \tilde{\sigma}_k(x_k) + \log w_g(x_k) - \mathcal{SG}(\log w_g(x_k)) + \log z^{pres}_{\phi_v(k)}$ where $w_g(\cdot)$ is a zero centered gaussian function and $\mathcal{SG}$ is the stop gradient operation. By adding $\log w_g(x_k) - \mathcal{SG}(\log w_g(x_k))$, we encourage the $z^{where}_{\phi_v(k)}$ to be set at the object center with the value of the weighted density unchanged. Weighted by $z^{pres}_{\phi_v(k)}$, non-existent components are turned off. We then compute the normalized density as $\bar{\sigma}_k(x_k) = \frac{\hat{\sigma}_k(x_k)^2}{\sum_{i=0}^{K} \hat{\sigma}_i(x_i)}$. Note that $\sum_k \bar{\sigma}_k(x_k) \in [0, 1]$ allowing us to represent concrete object or void space. The final NeRF density at point $x$ is given by $\sigma(x) = \sigma_{max} \cdot \sum_{k=0}^{K} \bar{\sigma}_k(x_k)$ with $\sigma_{max}$ being the maximum NeRF density of our choice.

We do not down-weight air sample densities by $w_g(\cdot)$ and $z^{pres}_k$ to force the model to learn the scene voidness at all ranges. At this point, the decoding/reconstruction is complete and we can compute the reconstruction likelihood.

**Refinement Network.** Following the amortized variational inference literature (Greff et al., 2019; Li et al., 2020), a refinement network takes as input latent variables from the previous step and a set of auxiliary data and outputs the updated proposal distributions (Eq. 4). The auxiliary data include observation, previous reconstruction and observation likelihood. For each input view, the above reconstruct-and-refine process is executed $L$ times, in parallel for all components. Notably, our objective function enforces the refinement process to preserve object identities across views without any hard-coded heuristics (detailed in Appendix B). Detailed network structure, auxiliary input specifications and algorithmic summary are presented in Appendices E and A.

**Cognitive Map.** A cognitive map maintains all discovered objects with a list where the $i^{th}$ entry stores $\lambda_i$ for object $i$ and interacts with the inference model through two functions:

Query: Given a camera $v$ specified in the global coordinate system, we retrieve $\{(i, \lambda_i)\}_{i=1}^{K}$ for $K$ objects whose locations are inside the field of view for camera $v$, from the cognitive map. To this end, according to the extrinsic parameters of $v$, $\mu^{where}$ for all objects in the cognitive map are projected from the global coordinate system into the camera coordinate system of $v$ while other parameters are kept fixed. If less than $K$ objects are located in the current view, priors of latent variables are filled in with pseudo index $i = -1$.

Registration: Given a camera $v$, the set $\{(i, \lambda_i)\}_{i \in \{\phi_v(1)...\phi_v(K)\}}$ can be registered into a cognitive map. Objects with $q(z^{pres} = 1) < 0.5$, will be discarded since they are deemed non-existent. As the reverse process of a query, all $\mu^{where}$ will then be projected into the global coordinate system. Then, newly discovered objects (identified by $i = -1$) are appended to the list and the corresponding entry indices become their global object id.

**Per-Object NeRF Finetuning.** To achieve higher reconstruction quality, we associate each object representation with a per-object NeRF (no weight sharing) in the cognitive map. We initialize the per-object NeRF decoder as that of the SOOC3D NeRF and then finetune it by maximizing the input view observation likelihood. Such non-weight sharing and training strategy for per-object NeRF can improve the reconstruction quality and preserve object identity naturally. For implementation, we fix $z^{where}$, binarize $z^{pres}$ with a threshold of 0.5 and treat view $z^{what}$ as part of the NeRF parameters.

## 4 EXPERIMENTS

**Dataset.** Datasets adopted in object-centric learning literature commonly focus on small scenes that can fit into camera FOV (Eslami et al., 2018; Johnson et al., 2017; Engelcke et al., 2021; Yu et al., 2022). To evaluate the scalability of our approach, we construct a large-scale dataset termed *Unity dataset* mimicking the object room dataset (Eslami et al., 2018). The dataset contains scenes of three different scales termed as *small* (s), *medium* (m) and *large* (l). To assess the performance of our model when facing non-trivial geometries, we also build a *Blender* dataset where objects are randomly selected from a pool of indoor furniture. See Appendix D for dataset generation details.

**Metric.** We compute the mean-intersection-over-union (mIoU) score between the ground truth mask and the mask inferred from our NeRF representation for scene segmentation and per-pixel root-mean-square-error (RMSE) for the rendered RGB image and the depth for the scene reconstruction. We mask out all pixels with depth values larger than the clipping plane. We compute both mIoU and

RMSE for both input (I) and query (Q) views. We also evaluate the location prediction performance by measuring the average distance between the predicted object locations and the ground truth for each scene. See Appendix F for rendering equations of depth, segmentation masks, and uncertainty maps. Unless stated otherwise, the visualizations below are without per-object NeRF finetuning.

**Baseline.** We compare our method with MulMON (Li et al., 2020), the state-of-the-art multi-view 3D scene object-centric learning method with online inference ability. We adopt their official implementation and additionally add ground truth depth as input. We additionally discuss ObSuRF Stelzner et al. (2021), a single view inference model, in Appendix G to provide more insights.

### 4.1 SCALABLE OBJECT CENTRIC LEARNING

Test time scalability is crucial for deploying agents in an open environment. Below we identify three types of scalability desired for an object-centric learning model. Our experiments show that our model demonstrates strong test time scalability across all three types.

**Scene Scale.** In practice, the training scenes are normally of bounded scale but the deploying scenes are of varying sizes. Below, we test the generalization of our approach w.r.t a different scene scale. Specifically, we train the baseline and our model on both small (MulMON_small, Ours_small) and medium (MulMON_medium, Ours_medium) scenes and test them on different scene scales. For the baseline, we set the number of mixing components $M = 8, 12$, and 25 for small, medium and large scale scenes respectively, during both training and testing. For our model, we fix $K$ to 7 for all scene scales. The quantitative results are reported in Table 1. The results of our approach are obtained as the average of 5 runs with at most $\pm 0.01$, $\pm 0.02$, $\pm 0.02$ and $\pm 0.02$ variation for mIoU, RGB-RMSE, depth-RMSE and coordinate L-2 error respectively. MulMON achieves 0.612 mIoU when trained and tested on small scenes. This setup aligns with their assumption that all objects appear in all views. However, evaluated on medium and large-scale scenes, its performance drops significantly to below 0.2 in mIoU. The performance of MulMON is not improved after training on the medium scene.

Table 1: Quantitative results on scene segmentation and reconstruction.

| | | mIoU ↑ | | | RGB RMSE ↓ | | | depth RMSE ↓ | | | L2 coord. error ↓ | | |
|---|---|---|---|---|---|---|---|---|---|---|---|---|---|
| | | s | m | l | s | m | l | s | m | l | s | m | l |
| MulMON_small | I | 0.612 | 0.198 | 0.158 | **0.055** | 0.167 | 0.178 | | N/A | | | N/A | |
| | Q | 0.599 | 0.192 | 0.152 | 0.056 | 0.171 | 0.186 | | N/A | | | N/A | |
| MulMON_medium | I | 0.371 | 0.225 | 0.141 | 0.102 | 0.141 | 0.144 | | N/A | | | N/A | |
| | Q | 0.365 | 0.221 | 0.136 | 0.103 | 0.149 | 0.159 | | N/A | | | N/A | |
| Ours_small | I | **0.763** | 0.721 | 0.694 | 0.074 | 0.107 | 0.141 | **0.516** | 0.680 | 0.793 | | N/A | |
| | Q | 0.761 | 0.713 | 0.690 | 0.073 | 0.109 | 0.149 | 0.517 | 0.691 | 0.805 | **0.068** | 0.170 | 0.192 |
| Ours_medium | I | 0.710 | **0.756** | **0.761** | 0.075 | **0.098** | **0.091** | 0.617 | **0.650** | **0.634** | | N/A | |
| | Q | 0.703 | 0.751 | 0.757 | 0.073 | 0.099 | 0.092 | 0.619 | 0.652 | 0.640 | 0.099 | **0.117** | **0.111** |
| Ours_finetune | I | 0.903 | 0.897 | 0.910 | 0.016 | 0.018 | 0.021 | 0.472 | 0.620 | 0.615 | | N/A | |
| | Q | 0.892 | 0.874 | 0.889 | 0.019 | 0.023 | 0.025 | 0.498 | 0.630 | 0.624 | | N/A | |

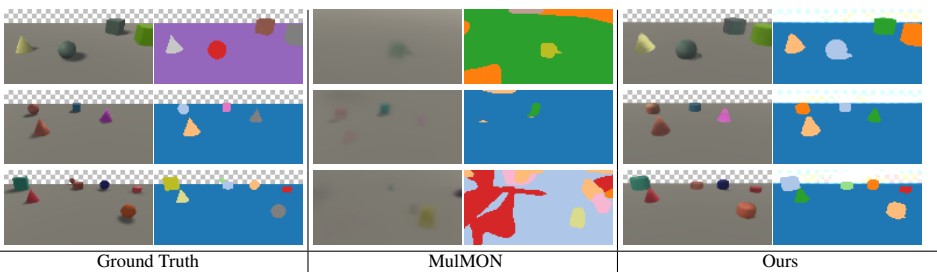

| Ground Truth | MulMON | Ours |

Figure 2: Qualitative comparison on query view synthesis in large scenes.

By contrast, our model, being agnostic to any global coordinate system, can scale to large scenes with a 0.07 mIoU performance drop when it is only trained on small scenes. After we train our model on medium scenes, the performance in both medium and large scenes improves (see Fig. 2 for qualitative comparison). The similar performances on the input and query views demonstrate that our model learned the view-invariant features resulting in robust rendering from all views. As shown in the last row of Table 1, after per-object finetuning, on the unity dataset, both instance mask prediction and appearance reconstruction improved significantly (qualitative results are shown in Appendix H).

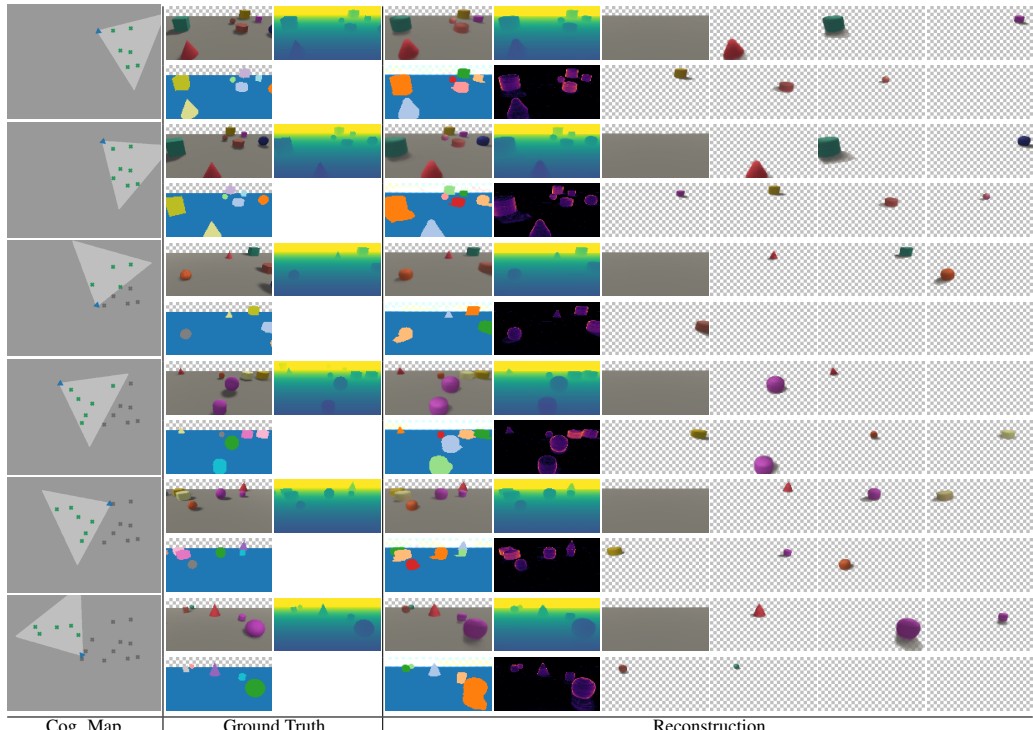

| Cog. Map | Ground Truth | Reconstruction |
|---|---|---|

Figure 3: Visualization of an online inference process from top to bottom. Each row corresponds to one inference step. The left column shows the evolution of the cognitive map. The camera pose of each step is marked by a blue triangle and the camera cone (visible area) is highlighted. Each object latent registered in the cognitive map is marked with an x and is greyed out if outside of view.

In Fig. 3 we visualize a 6-step online inference process. For each step, our model discovers new objects, registers them to the cognitive map and at the same time updates the latent variables. With a cognitive map as external memory, the inference process can be scaled to arbitrary scales.

**The Number of Object in View.** To demonstrate the scalability w.r.t the number of objects in the scene, we generate a set of 100 testing scenes, each of which consists of 10 views. Each view can capture 8 to 11 objects. We set the $K$ to be 11 and test our pre-trained model without any additional training. We then report that our model achieves on average **0.702** mIoU, and **0.092** for RGB-RMSE. Although it has never been trained on such dense views, our model can generalize well to dense 3D scenes without performance drops. Qualitative results on scene decomposition are shown in Fig. 4. Our method can decompose the images with dense objects and render each object with high quality.

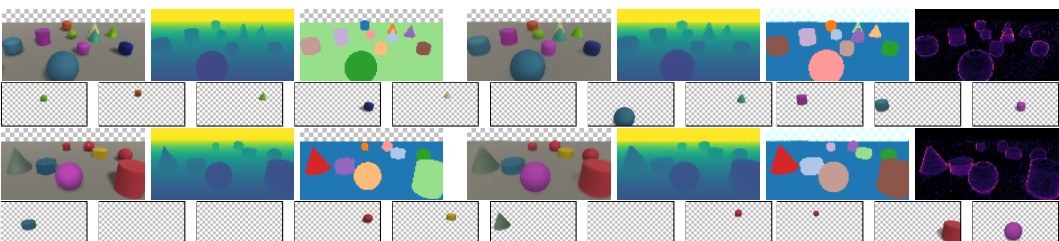

Figure 4: Demonstration of the model scalability at test time on the number of objects per view. In rows 1 and 3, from left to right are ground truth RGB, depth, instance mask, rendered RGB, rendered depth, rendered mask and uncertainty map. Rows 2 and 4 show the rendered individual objects.

**Number of Updates.** In practice, objects can be observed and updated arbitrarily frequently and the inferred scene representations should remain stable. However, during training, it is common that each

object latent can only be updated a fixed maximum number of times. In our experiments, one object latent can be updated 4 times at most during training.

To evaluate our model's scalability on object latent updates without additional training, we sample input views and repeat the inference process multiple times. After each update, we compute the induced KL Divergence and the RGB-RMSE for the rendered image. As shown in Fig. 5, our model shows test-time scalability to around 10 updates. To achieve stronger scalability, we finetune our model by traversing each scene 5 times consecutively. Similar to the truncated backpropagation through time (TBPTT), after each traverse, we update the model parameters and keep the contents of the cognitive map for the next traverse. As a result, the object representation remains stable under more than 30 updates during testing time. Fig. 6 visualizes the appearance changes under updates.

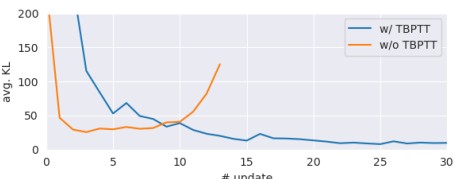 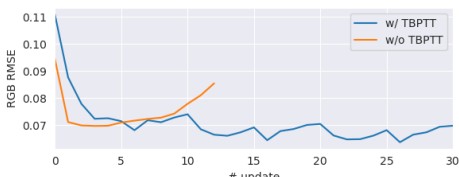

Figure 5: The per-update KL (left) and per-update RGB RMSE (right) with and without finetuning.

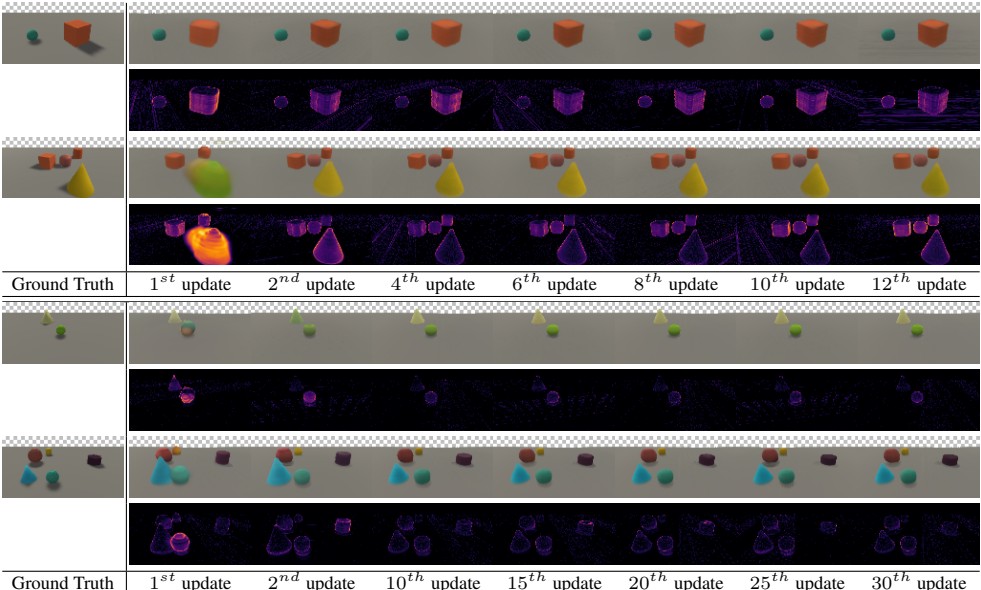

Figure 6: Visualization of scene changes under update before (top two sequences) and after (bottom two sequences) the TBPTT finetuning. Due to object blocking views, the first update may result in high uncertainty. Qualitatively, the appearance of objects is refined and remains stable after.

## 4.2 SCENE EDITING

In previous works, while object removal or creation can be achieved by adding or deleting object latent variables, moving objects commonly relies on latent variable traversal (Kabra et al., 2021; Burgess et al., 2019; Greff et al., 2019; Li et al., 2020). For our model, thanks to the explicit object pose estimation, scene editing can be achieved with a direct modification of the content of the cognitive map in a view-independent way. We use our model to infer the latents of a random small scene and save the resulting Cognitive map. In Fig. 7, we modify the registered coordinates of the latents in the Cognitive map by directly setting them to some designated coordinates to create different object arrangements. In Fig. 8, we traverse $z^{what}$ of the left-most object one dimension at a time, render the modified object and inspect the changes in the object appearance manually. We identified three feature dimensions with highly interpretable meanings, i.e. color, shape and size.

### 4.3 VISUALLY CHALLENGING SCENES AND PER-OBJECT FINETUNING

We test our model on the Blender dataset to evaluate the capability of handling more realistic scenes. We report that our model achieves **0.734 mIoU** and **0.072 RMSE** while MulMON achieves **0.489 mIoU** and **0.069 RMSE**. The experimental results show that our model is capable of handling challenging geometries with photorealistic observations. The metrics can be further improved by the per-object finetuning to **0.870 mIoU** and **0.031 RMSE**. The comparison between ground truth observation (GT), SOOC3D inference results (SOOC3D) and the per-object finetuning results (SOOC3D+) in Fig. 9 shows that the per-object finetuning process can capture the fine detail of objects leading to more accurate instance masks. See Appendix I for more qualitative results.

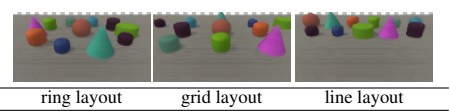

ring layout          grid layout          line layout

Figure 7: Demonstration of scene editing and rendering from our learned object representations. Note that all three images are rendered by the same set of learned objects representations.



color          shape          size

Figure 8: Latent variable traversal.

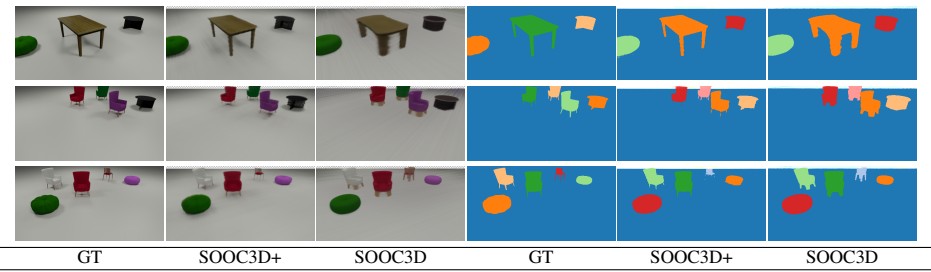

GT          SOOC3D+          SOOC3D          GT          SOOC3D+          SOOC3D

Figure 9: Qualitative comparison before and after per-object finetuning.

## 5 LIMITATION

We observe that our model may fail to model objects located on the boundaries of camera visual cones. This challenge is rarely discussed in previous 3D aware object-centric learning literature since they either process all frames at once (Kabra et al., 2021; Henderson & Lampert, 2020) or the scenes are not large enough to trigger the problem (Li et al., 2020; Stelzner et al., 2021; Yu et al., 2022). In a cognitive map, objects are abstracted as points. A large object can be partially observed while its predicted centers are still outside the current view. As a result, the object is not retrieved from the cognitive map. We anticipate that this problem can be handled by predicting a bounding box for each object. Then the cognitive map should retrieve any object whose bounding box intersects with the visual cone for the current view. We leave the bounding box prediction as our future work.

## 6 CONCLUSION AND FUTURE WORK

We propose a framework for unsupervised 3D object-centric learning for handling scenes of large scale and a varying number of objects in the scene. We introduced factorized latent learning which separates the camera pose and view-invariant object latents. Our object-compositional nerf allows the learning of 3D representation in the object camera coordinate system. The cognitive map ensures the framework keeps all the detected objects. Our learned view-invariant 3D object representation can potentially be applied in the SLAM system or the robotic object manipulation of object representation. Our model cannot handle dynamic objects in the scene which is left as our future work.

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

## A ALGORITHMS

---

**Algorithm 1:** SOOC3D Inference

**Input:** a cognitive map $\mathcal{M}$.

1 **begin**

    `/* iterate over views, `$T$` can be unbounded during test time */`

2    **for** $t = 1, 2, ..., T$ **do**

3        Receive posed observation $(o_{v_t}, v_t)$;

4        $\lambda_t^{scene} = Encode(o_{v_t})$;

5        $z_t^{scene} \sim q_{\lambda_t^{scene}}$ ;                `// Gaussian distribution`

        `/* `$\phi_{v_t}(i)$` is the global index of the `$i^{th}$` object in view `$v_t$` */`

6        $\{(\lambda_{i,t}^0, \phi_{v_t}(i))\}_{i=1:K} = Query(\mathcal{M}, v_t), \lambda_{i,t}^0 = \{\lambda_{i,t}^{what,0}, \lambda_{i,t}^{where,0}, \lambda_{i,t}^{pres,0}\}$;

7        **for** $l = 0, 1, \ldots, L$ **do**

            `/* Executed in parallel                              */`

8            **for** $i = 1, \ldots K$ **do**

9                $z_{i,t}^{what,l} \sim q_{\lambda_{i,t}^{what,l}}$ ;           `// Gaussian distribution`

10                $z_{i,t}^{where,l} \sim q_{\lambda_{i,t}^{where,l}}$ ;        `// Gaussian distribution`

11                $z_{i,t}^{pres,l} \sim q_{\lambda_{i,t}^{pres,l}}$ ;         `// Bernoulli distribution`

12                Compute $\mathcal{D}_{KL,i}^{what} = \mathcal{D}_{KL}(q_{\lambda_{i,t}^{what,l}} || q_{\lambda_{i,t}^{what,0}})$;

13                Compute $\mathcal{D}_{KL,i}^{where} = \mathcal{D}_{KL}(q_{\lambda_{i,t}^{where,l}} || q_{\lambda_{i,t}^{where,0}})$;

14                Compute $\mathcal{D}_{KL,i}^{pres} = \mathcal{D}_{KL}(q_{\lambda_{i,t}^{pres,l}} || q_{\lambda_{i,t}^{pres,0}})$;

15            **end**

16            $\mathcal{D}_{KL} = \sum_{i=1}^{K}(\mathcal{D}_{KL,i}^{what} + \mathcal{D}_{KL,i}^{where}) \times z_{i,t}^{pres,l} + \mathcal{D}_{KL,i}^{pres}$;

17            $\mathcal{L}_t^l = -\log p(o_{v_t} | z_{1:k,t}^{what,l}, z_{1:k,t}^{where,l}, z_{1:k,t}^{pres,l}, z_t^{scene}) + \mathcal{D}_{KL}$;

            `/* `**a**` is the auxiliary input specified in Appendix E.   */`

18            $\lambda_{i,t}^{l+1} = \lambda_{i,t}^l + f_\vartheta(z_{i,t}^{what,l}, z_{i,t}^{where,l}, z_{i,t}^{pres,l}, o_{v_t}, v_t, \mathbf{a})$;

19        **end**

20        $Register(\{(\lambda_{i,t}^{L+1}, \phi_{v_t}(i))\}_{i=1,\ldots,K}, \mathcal{M}, v_t)$

21    **end**

22 **end**

23 **return** *Cognitive Map $\mathcal{M}$ storing the object-centric scene representation.*

---

---

**Algorithm 2:** Cognitive Map Register

---

**Input:** a cognitive map $\mathcal{M}$, latents with global index $\{(\lambda_i, \phi(i))\}_{i=1,...,K}$, camera pose $v$.

1 **begin**
2     **for** $i = 1, 2, ..., K$ **do**
3         $\lambda_i^{what}, \lambda_i^{what}, \lambda_i^{what} = \lambda_i$;
4         $\mu_i^{where}, \sigma_i^{where} = \lambda_i^{where}$;
5         $\mu_i^{where} \leftarrow ProjectIntoGlobalCoordinate(\mu_i^{where}, v)$;
6         **if** $\phi(i) = -1$ *and* $q_{\lambda_i^{pres}}(z^{pres} = 1) > 0.5$ **then**
                `/* Object newly detected in this view          */`
7             $\phi(i) \rightarrow len(\mathcal{M})$;
8             $\mathcal{M}.append(\lambda_i)$;
9         **else**
                `/* Object detected by previous views           */`
10            $\mathcal{M}[\phi(i)] \leftarrow \lambda_i$;
11         **end**
12     **end**
13 **end**
14 **return** *Cognitive Map $\mathcal{M}$ storing the object-centric scene representation.*

---

**Algorithm 3:** Cognitive Map Query

---

**Input:** a cognitive map $\mathcal{M}$, a camera pose $v$, global prior $\lambda_{prior}$.

1 **begin**
2     initialize empty list $L = []$;
    `/* Find all existing object in view                  */`
3     **for** $i = 0, 1, 2, ..., len(\mathcal{M}) - 1$ **do**
4         $\lambda_i^{what}, \lambda_i^{what}, \lambda_i^{what} = \lambda_i$;
5         $\mu_i^{where}, \sigma_i^{where} = \lambda_i^{where}$;
6         **if** $\mu_i^{where}$ *in the FOV of* $v$ *and* $q_{\lambda_i^{pres}}(z^{pres} = 1) > 0.5$ **then**
7             $\mu_i^{where} \leftarrow ProjectIntoGlobalCoordinate(\mu_i^{where}, v)$;
8             $L.append((\lambda_i, i))$;
9         **end**
10     **end**
    `/* Rank latents with the probability of existence and take`
       `the top K latents                                */`
11     $L \leftarrow TopK(L, q_{\lambda_i^{pres}}(z^{pres} = 1))$
12     **while** $len(I) < K$ **do**
13         $L.append((\lambda_{prior}, -1))$
14     **end**
15 **end**
16 **return** $L$ *containing $K$ queried object latents.*

# B ELBO OBJECTIVE.

**Object Centric NeRF Likelihood.** In our formulation, each pixel in one image emits a single ray. Thus, with known intrinsic parameters of the camera, ray directions can be computed from pixel coordinates. Each ray is associated with two observables, i.e. the color and the depth of the hit point of the ray. Following common practice, we assume the likelihood of color and depth are independent given latents $Z_{v_t}$.

For a single-component NeRF, the color returned by a ray $r(\cdot)$ originated from $x_0$ pointing towards direction $e$, i.e, $r(t) = x_0 + et$, in a neural radiance filed is defined by the rendering equation

$$C(r) = \int_0^\infty T(t)\sigma(r(t))c(r(t), e)dt, \tag{6}$$

where $\sigma(r(t))$ and $c(r(t), e)$ are NeRF density and color at point $r(t)$ separately, $T(t) = \exp\left(-\int_0^t \sigma\left(r\left(t'\right)\right)dt'\right)$ is the transmittance term (Mildenhall et al., 2021). Under mild assumptions, the probability density that the observed colors originate at the depth $t$ via the ray $r(\cdot)$ is given by $p(t, r) = \sigma(r(t))T(t)$ (Stelzner et al., 2021). An unbiased estimator is given by

$$\log p_\theta(t, r|\mathcal{Z}_v) = \log \sigma_\theta(r(t)|\mathcal{Z}_v) - \mathbb{E}_{t' \sim q(\cdot)}\left[\sigma_\theta\left(r\left(t'|\mathcal{Z}_v\right)\right)/q\left(t'\right)\right], \tag{7}$$

where $q(\cdot)$ is a proposal distribution with support $[0, t]$, $\theta$ is the parameters of a NeRF decoder.

By replacing $t$ in Eq. 7 with depth $d_v^j$, we can compute the likelihood $p(d_v^j|\cdot)$ of the depth $d_v^j$ of pixel $j$. The color likelihood of pixel $j$ is given by $p(c_v^j|\cdot) = \mathcal{N}(c_v^j|\hat{c}_v^j, \sigma_c)$, where $\hat{c}_v^j$ is the predicted color and $\sigma_c$ is a constant variance. Thus, the per-pixel log likelihood is decomposed as $\log p(o_v^j|\mathcal{Z}_v) = \log p(d_v^j|\mathcal{Z}_v) + \log p(c_v^j|\mathcal{Z}_v)$. For each pixel, we only need to evaluate the NeRF at the depth $d_v^j$, and a random point sampled from $q(\cdot)$. Points beyond a fixed clipping depth are deemed unreliable and are discarded.

Under the object-centric learning setup, the object-compositional NeRF is a composition of $K + 1$ NeRFs. The NeRF density $\sigma(r(t)) = \sum_{k=0}^K \sigma_k(r(t))$ is the sum of the density of each component. The color $\hat{c}(r(t)) = \sum_{k=0}^K \frac{\sigma_k(r(t))}{\sigma(r(t))}\hat{c}_k(r(t), e)$ is now a weighted sum of individual components.

Following common practice, we assume the likelihood of rays is independent given object latent. Then the joint likelihood $p(o_{v_t}|Z_{v_t})$ in Eq. 5 is the product of the likelihood of all rays. In practice, for each view/image, instead of using all rays/pixels, for efficiency reasons, we only sample a set of rays/pixels to compute the likelihood. By plugging the likelihood function into the Eq. 5, we get the ELBO that SOOC3D is trained to maximize.

The KL term $\mathcal{L}_{v_t}^{kl}$ in Eq. 5 is crucial to the emerging of view-invariant object-centric representations. For each refinement iteration $l$ at time step $t$, we compute the KL term as

$$z_{i,t}^{pres,l}\left[\mathcal{D}_{KL}(q_{\lambda_{i,t}^{what,l}}||q_{\lambda_{i,t}^{what,0}}) + \mathcal{D}_{KL}(q_{\lambda_{i,t}^{where,l}}||q_{\lambda_{i,t}^{where,0}})\right] + \mathcal{D}_{KL}(q_{\lambda_{i,t}^{pres,l}}||q_{\lambda_{i,t}^{pres,0}}). \tag{8}$$

Recall that, at the beginning of the inference process the cognitive map is empty and, instead, global priors will be retrieved from the cognitive map. The global prior of $z^{what}$ and $z^{where}$ takes the form of zero-mean isotropic Gaussian. To encourage the rejection of empty component, the global prior of $z^{pres}$ is a Bernoulli distribution with $p(z^{pres} = 1) \approx 0$. We further encourage the rejection of empty components by weighing $\mathcal{D}_{KL}(q_{\lambda_{i,t}^{what,l}}||q_{\lambda_{i,t}^{what,0}})$ and $\mathcal{D}_{KL}(q_{\lambda_{i,t}^{where,l}}||q_{\lambda_{i,t}^{where,0}})$ with $z^{pres}$. As a consequence, rejected components will not induce KL penalty.

Crucially, the KL term encourages object latents to remain constant across views. One local optimum for models maintaining a dynamic set of object representations is to discard all previously detected objects and re-discover all objects in the current view (Crawford & Pineau, 2020). Instead of learning view-invariant representations, the discover-reconstruct-discard mode achieves a high reconstruction likelihood by only extracting view-dependent representations. The KL term in our objective discourages such local optimum since re-discovering an object induces a high KL divergence between the global prior and the posterior.

Another common concern is that when one object is temporally blocked by other objects, it is excluded from the likelihood computation directly and can be discarded without changing the observation likelihood. The KL term between $z^{pres}$ preserves the existence of such blocked objects.

The refinement network takes into account both object location $z^{where}$ and appearance information $z^{what}$. By minimizing the KL term between each update, we discourage any abrupt changes to our latents. The KL term between $z^{where}$ enforces better handling of multiple objects with the same appearance. Thus, exchanging the location of two objects will lead to a large KL loss. Note that in practice two rigid bodies cannot occupy the same space. If two objects share the same space and have the same appearance information, our model will learn to reject one of them by setting the corresponding $z^{pres}$ to 0 since rejecting one will not decrease the likelihood.

## C  TRAINING

In this section, we describe our training pipeline. For each scene in the training set, SOOC3D first takes a set of $T$ views sequentially as input and detects and refines object representations. Then the set of object representations refined through $T$ views is used for query view reconstruction. The likelihood of $Q$ query views is computed before we get the ELBO as specified in Eq.5.

While the cognitive map is a key factor to scalability, it also makes end-to-end training unstable. As introduced in the main paper, any object with $p(z^{pres} = 1) < 0.5$ will be rejected during cognitive map registration and objects that are outside the current view will not be retrieved. Objects can be constantly dropped during the registration and query process at the beginning of the training stage introducing a large variance.

To overcome this issue, we resort to curriculum learning and pre-train our model on small scenes that can fit into camera FOV (detailed below). During registration, all $K$ object latent distributions are kept regardless of their $z^{pres}$ value. Similarly, the query function skips the in-view check and returns all registered object latents. Thus, for each scene, we will have the same set of $K$ object throughout.

We stop the pre-training at $20,000$ iterations when our model can stably predict object locations and $z^{pres}$ value. Then we train our model on scenes with the full cognitive map functionality.

## D  DATASETS

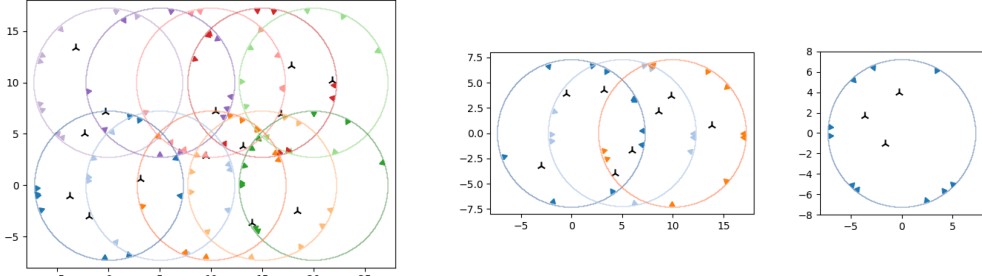

Figure 10: Scene setup example. From left to right is the large, medium and small scene. Objects are marked in black and cameras are marked with colored triangles. Cameras are uniformly randomly sampled from camera rings (color circles).

To evaluate the scalability of our approach, we construct a large-scale dataset with Unity3D (Juliani et al., 2020) termed *Unity dataset*. The dataset contains scenes of three different scales termed as *small*, *medium* and *large* scenes used for evaluating the scalability of our approach. For small and medium scenes we generate $40,000$ scenes for training and $1,000$ scenes for testing. For large scenes, we only generate $1,000$ scenes for testing.

For each *small* scene, $2 \sim 5$ objects are randomly placed in a square of 5 unit width. 10 cameras of resolution $128 \times 64$ are randomly sampled on a ring surrounding the area. A *medium* scene is twice the size of a small scene where $3 \sim 10$ objects are randomly placed in a 5 unit by 10 unit region. In particular, we treat each medium scene as the overlapping of three square areas, each of which is of 5 unit width. For each area, we spawn 10 cameras in a ring layout. Note that the size of the *medium*

scene is now too large to fit into a camera FOV. Different cameras in a medium scene may capture a different set of objects. In a large scene, $12 \sim 24$ objects randomly spawn in a 10 unit by 15 unit area, which is 3 times larger than the medium scenes. In terms of camera setup, we decompose the large area into 10 regions, each of which is a square area of 5 unit width. 10 cameras are set similarly as above. Thus for a large scene, we have in total 100 views available. The camera ring layouts are illustrated in Fig. 10.

During training, for each small scene, we randomly sample 3 input views and 2 query views. For each medium scene, in each area, we randomly sample 2 input views and 1 query view. For testing, for small and medium scenes, we keep the same number of input views and use the rest as query views. For large scenes, we randomly sample 2 input views in each area. A large-scene inference visualization is shown in Appendix H.

The *Blender dataset* is built in Blender with raytracing renderer. The object and camera arrangements follow the *small* scene setup of the *Unity dataset*. Each scene in the *Blender dataset* contains $3 - 7$ objects of non-trivial structures. The training set contains 10K small scenes, each of which contains 10 views. The training procedure is identical to the one described above. Similarly, during training, for each scene, we randomly sample 3 input views and 2 query views. The Blender dataset contains $10,000$ training scenes and 200 testing scenes.

Unlike the Unity dataset, for the Blender dataset, we render each view in $512 \times 256$ resolution. During training, we downsample observations to $128 \times 64$. For per-object finetuning, we use the original resolution.

## E  NETWORK STRUCTURE AND HYPERPARAMETERS

Auxiliary input: We group the auxiliary input into three different bundles.

| bundle | quantity | note |
|---|---|---|
| data bundle | $o_t$ | RGB and depth observation |
|  | depth mask | mask indicate whether the depth is beyond clipping plane |
|  | $e$ | ray direction in object local coordinate system |
|  | $x_k$ | object local coordinate of surface sample |
| latent bundle | $\lambda_i$ | object latent distribution |
|  | $d\mathcal{L}_{\lambda_i}$ | grad. of $\lambda_i$ w.r.t loss |
| inference bundle | $\log p(d)$ | depth likelihood of the composed NeRF |
|  | $\hat{\sigma}$ | predicted surface sample density |
|  | $\hat{\sigma}_k$ | predicted surface sample density of the $k^{th}$ NeRF component |
|  | $d\mathcal{L}_{\hat{\sigma}}$ | grad. of $\hat{\sigma}$ w.r.t loss |
|  | $\log p(c)$ | color likelihood of the composed NeRF |
|  | $\log p_k(c)$ | color likelihood of the $k^{th}$ NeRF component |
|  | $\log p_{/k}(c)$ | leave-one-out color likelihood |
|  | $\hat{c}$ | predicted color |
|  | $\hat{c}_k$ | predicted color of $k^{th}$ NeRF component |
|  | $d\mathcal{L}_{\hat{c}_k}$ | grad. of $\hat{c}_k$ w.r.t loss |

Hyperparamter:

| quantity | note | value |
|---|---|---|
| $lr$ | learning rate | $1e^{-4}$ |
| $\sigma_{max}$ | the maximum NeRF density | 5 |
| $\sigma_c$ | color prediction var. | 0.2 |
| L | the number of AVI iteration | 3 |
| R | the number of sampled ray | 2048 |
|  | observation resolution | $64 \times 128$ |
| D | $z^{what}$ dimension | 64 |

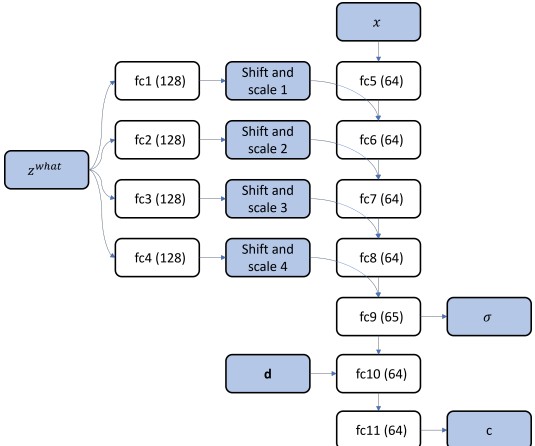

Figure 11: NeRF decoder structure. We specify the output dim for each fully connected layer. fc1-4 has no activation function. The rest fully connected layer uses leaky relu. $x$ is the harmonic representation of point coordinates. $\sigma$ and $c$ are the predicted raw density and color. $e$ is the ray direction.

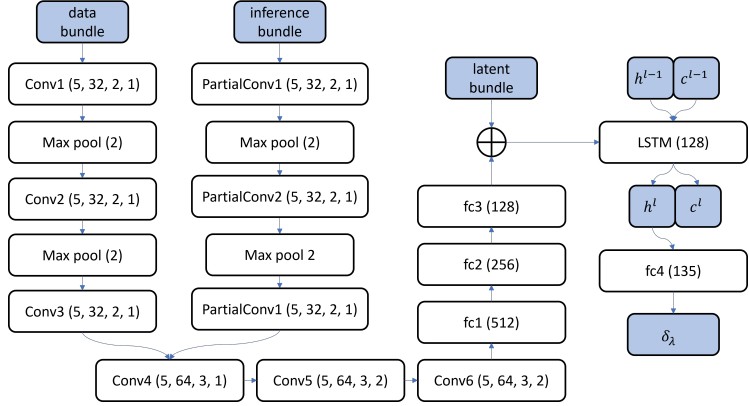

Figure 12: Refinement network structure. For convolution and partial convolution layer we specify (kernel size, output channel, padding, stride). For LSTM we specify the dimension of its hidden state. It produces the additive delta of $\lambda_i$.

## F    RENDERING EQUATION FOR DEPTH, MASK AND UNCERTAINTY

To render per-component RGB, we apply the render equation Eq. 9 on individual NeRF components to get the color. The transparency is rendered using equation Eq. 10. The returned transparency always fails in the interval $[0, 1]$.

$$C_k(r) = \int_0^\infty T_k(t)\sigma_k(r(t))c_k(r(t), e)dt. \tag{9}$$

$$A_k(r) = \int_0^\infty T_k(t)\sigma_k(r(t))dt. \tag{10}$$

Finally, we lay $C_k(r)$ on a checkerboard background with transparency $A_k(r)$. The Eq.11 computes the contribution made by each component to the rendering of the composed NeRF.

$$K_k(r) = \int_0^\infty T(t)\sigma_k(r(t))dt. \tag{11}$$

The segmentation mask is obtained by $\arg\max_k(K_k(r))$. Eq. 12 gives the rendered depth of a view.

$$D(r) = \int_0^\infty T(t)\sigma(r(t))tdt. \tag{12}$$

For the uncertainty map, we interpret the normalized raw density detailed in Sec.3.2 as the probability that a point in space is occupied by matters. The view uncertainty is computed as by Eq. 13 where $\mathcal{H}$ is the entropy function.

$$H(r) = \int_0^\infty T(t)\sigma(r(t))\mathcal{H}(\sigma(r(t)))dt. \tag{13}$$

# G  ADDITIONAL BASELINE.

In this section, we discuss ObSuRFStelzner et al. (2021) as an additional baseline. ObSuRF is a 3D scene object-centric learning model based on the slot attention Locatello et al. (2020) mechanism. ObSuRF relies on a global coordinate system and by design only takes observations from one view as input. Thus, ObSuRF is not directly comparable to our method.

Since slot attention is an iterative object-centric learning algorithm shown effective on 2D, 3D and video object-centric learning tasks (Stelzner et al., 2021; Locatello et al., 2020; Kipf et al., 2021). We attempted to replace the amortized variational inference process in our model with slot attention. The slot attention model structures are identical to the one used in ObSuRF and we decompose each slot into $z_i^{where}, z_i^{pres}, z_i^{what}$ for each object. We further allow slots to be carried over to the next frames (with projected $z^{where}$) for multi-view online updating. The training likelihood function remains unchanged and the KL term is replaced with $L2$ distance between latents.

After training the slot attention-based model in small scenes, we match the ground truth object identities with the predicted object identities after each input view. We report that the slot attention module only preserves **27**% object identities on average per update, while our full model equipped with amortized variational inference preserves identities all the time. Further investigation reveals that during subsequent slot attention iteration, the identities, as well as representations of slots are discarded and assigned.

The slot attention can be interpreted as a clustering algorithm operating on feature vectors extracted from observations. We conjecture that the feature vectors are view-dependent and the clustering mechanism fails to convert view-dependent features into view-independent object representations. Thus, compared with a multi-view setup, slot attention is better suited for continuous frame processing where the correlation between consecutive frames is large.

To compare with ObSuRF, we downgrade our model to the single view inference mode by setting the number of input views and the number of query views to be one and train on the CLEVR3D and the MultiShapeNet dataset Stelzner et al. (2021).

Table 2: Quantitative results on CLEVR3D and MultiShapeNet dataset. ObSuRF performance is obtained from the original paper.

|  | CLEVR-3D | | | | MultiShapeNet | | | |
|  | MSE $\times 10^3$ | Fg-Depth-MSE | Fg-ARI | ARI | MSE $\times 10^3$ | Fg-Depth-MSE | Fg-ARI | ARI |
|---|---|---|---|---|---|---|---|---|
| ObSuRF | **0.78** | **0.10** | 95.7 | **94.6** | **1.91** | **3.44** | 81.4 | **64.1** |
| Ours | 0.84 | 0.12 | **96.2** | 92.2 | 2.04 | 3.81 | **83.1** | 61.4 |

Our model achieves performance on par with ObSuRF. Note that our model is not designed for the single view scenario.

# H    ADDITIONAL RESULTS ON UNITY DATASET

Below we show per-object NeRF finetuning results on the Unity dataset. We show the ground truth data (GT), SOOC3D inference results (SOOC3D) as well as the results of 5000 iteration per-object NeRF finetuning (SOOC3D+). While SOOC3D can capture the structure of objects, the finetuning process recovers more details like edges or shadings. Most importantly, the object identities are preserved throughout the finetuning process.

Note that per-object NeRFs are also built on the object local coordinate system. Thus, one can finetune an unboundedly large scene and register each per-object NeRF into the cognitive map to achieve **unsupervised unboundedly scalable high-quality object-compositional NeRF reconstruction**.

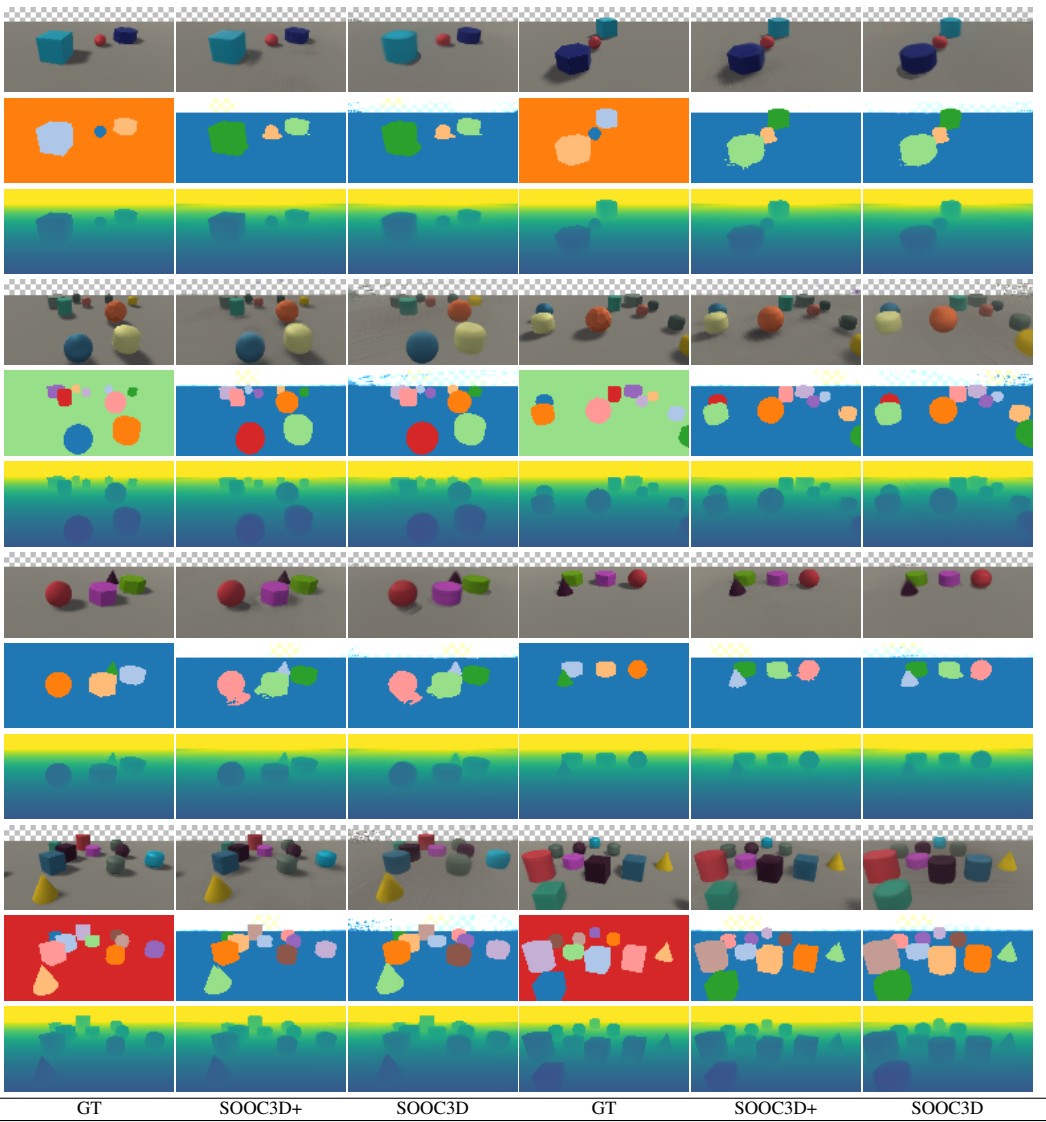

Figure 13: Qualitative comparison before and after per-object finetuning.

To demonstrate the scene scale scalability, below we show a long inference sequence (without per-object finetuning) in a large scene in Fig. 14 and Fig. 15.

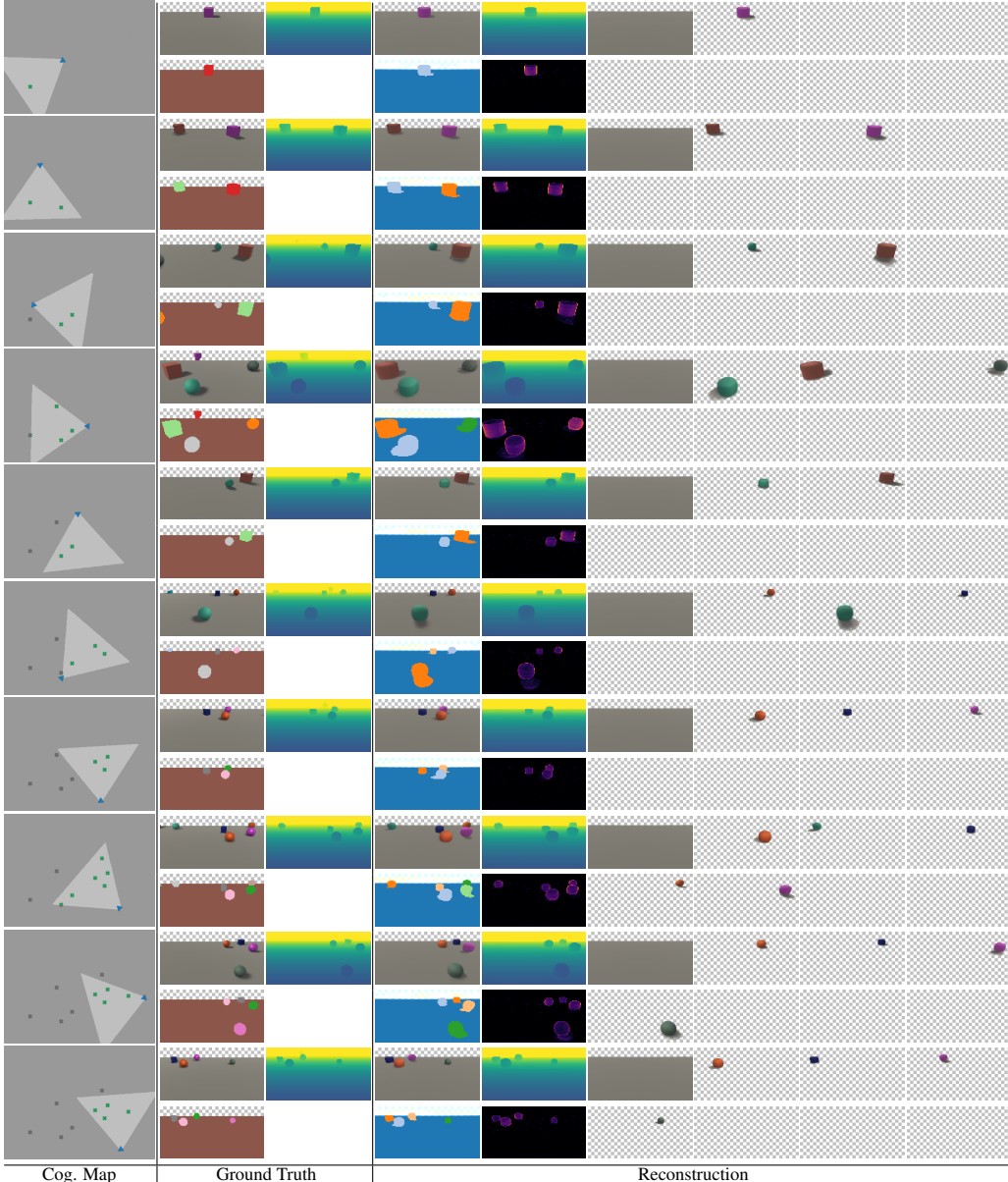

Figure 14: Visualization of steps 1-10 in a 20-step inference trajectory.

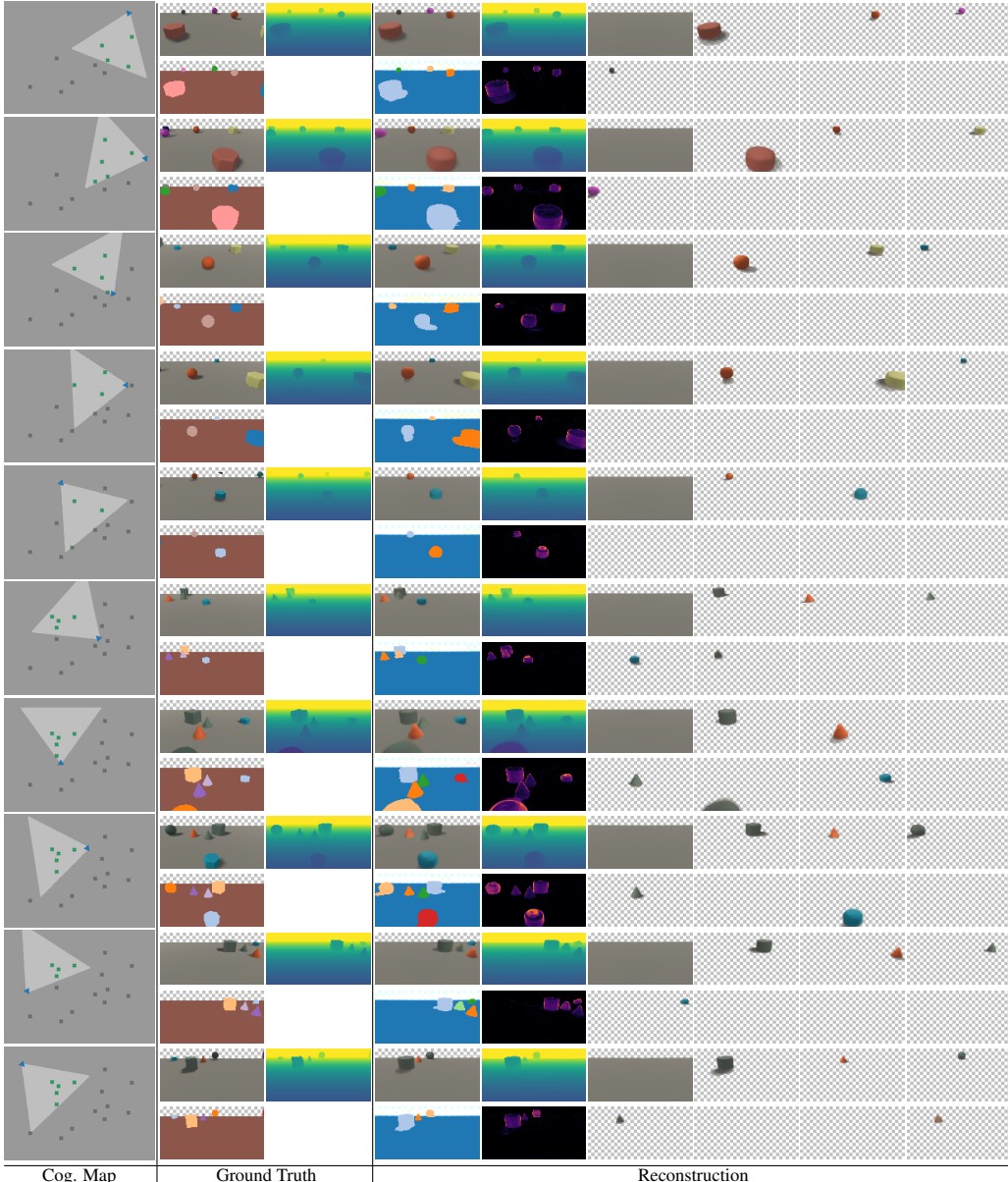

Figure 15: Visualization of steps 11-20 in a 20-step inference trajectory.

# I  ADDITIONAL RESULTS ON BLENDER DATASET

Fig. 16 shows the qualitative comparison between our methods and MulMON Li et al. (2020) baseline. Both our model (SOOC3D) and MulMON are trained with low-resolution observation ($128 \times 64$). As a NeRF-based method, our model can render images of arbitrary resolution, which allows us to finetune the per-object NeRF with higher resolution observation ($512 \times 256$).

When facing non-trivial geometries of varying sizes, our model can capture object structures accurately with correct pose prediction. Notably, after per-object NeRF finetuning, fine details are recovered leading to more accurate RGB, depth as well as instance mask prediction. While thin object parts like the legs of round tables are not reflected on the rendered instance masks (the main source of mIoU error) due to NeRF sampling interval length, we'd like to point out that they are correctly modeled by our SOOC3D+ as shown in their RGB reconstructions.

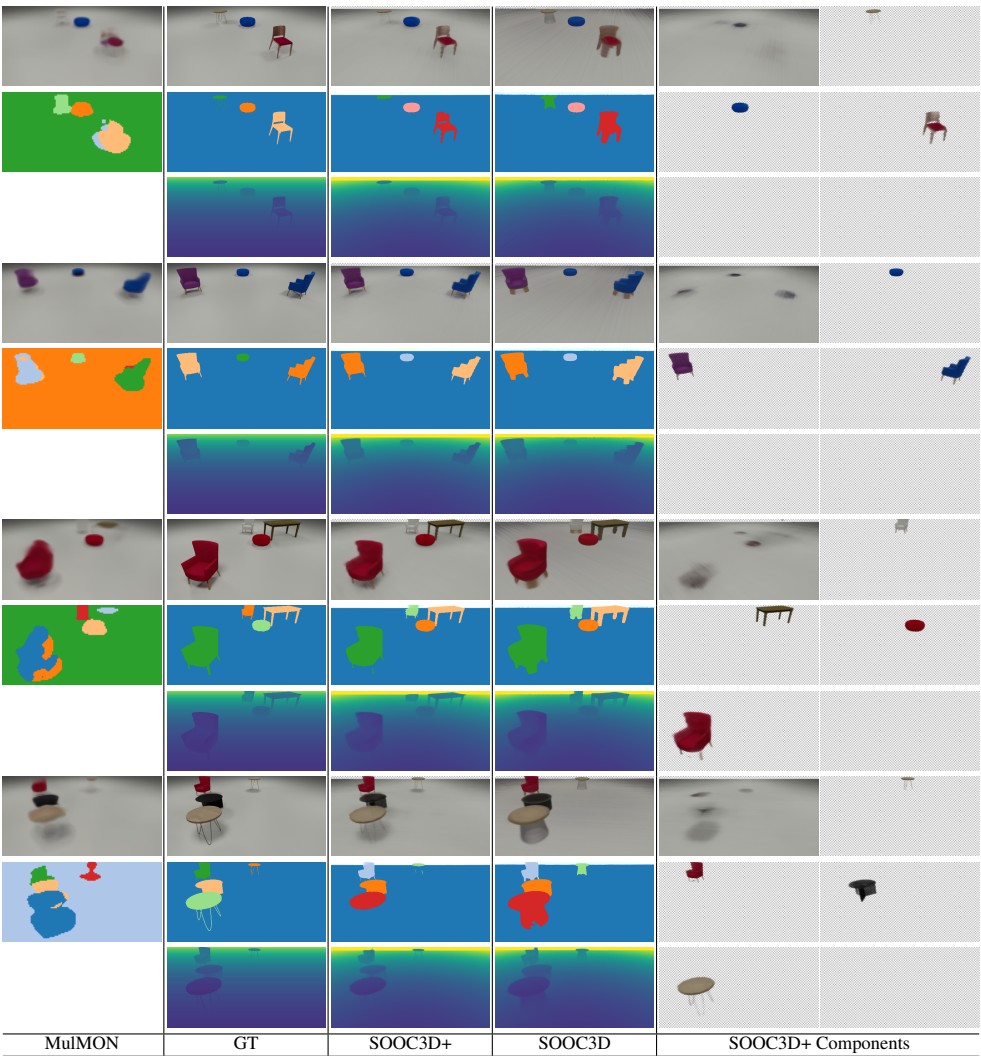

Figure 16: Qualitative comparison between the inference results of baseline (MulMON), the inference results of our model (SOOC3D) and our per-object finetuning results (SOOC3D+).

Below we show more SOOC3D+ results.

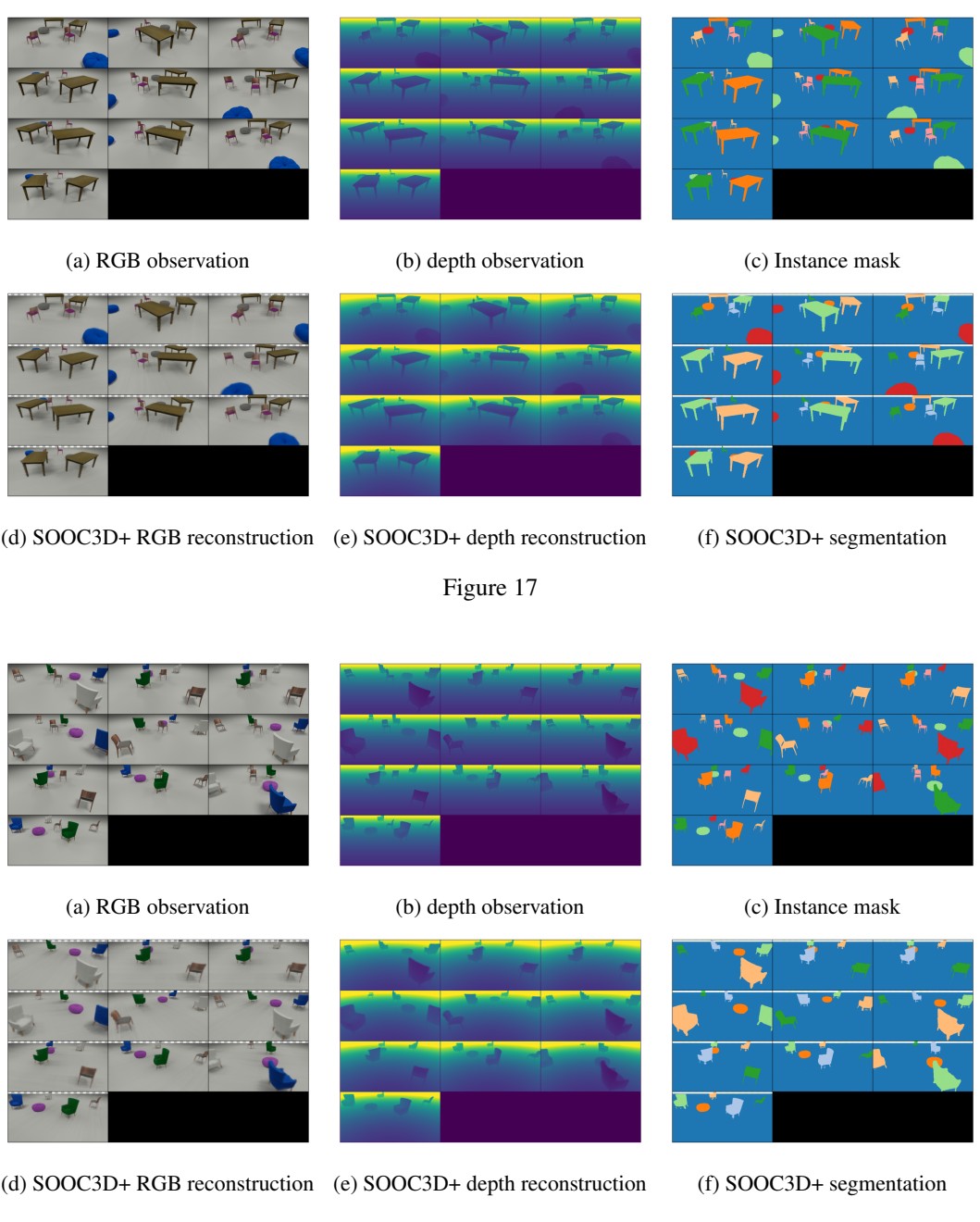

(a) RGB observation (b) depth observation (c) Instance mask

(d) SOOC3D+ RGB reconstruction (e) SOOC3D+ depth reconstruction (f) SOOC3D+ segmentation

Figure 17

(a) RGB observation (b) depth observation (c) Instance mask

(d) SOOC3D+ RGB reconstruction (e) SOOC3D+ depth reconstruction (f) SOOC3D+ segmentation

Figure 18

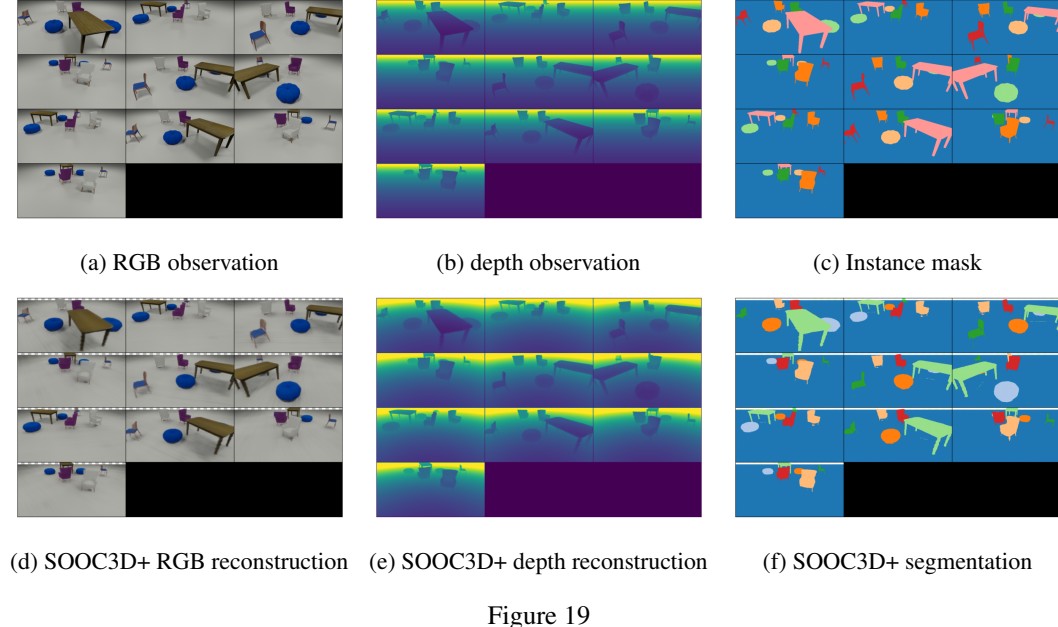

(a) RGB observation      (b) depth observation      (c) Instance mask

(d) SOOC3D+ RGB reconstruction    (e) SOOC3D+ depth reconstruction    (f) SOOC3D+ segmentation

Figure 19

