# OpenReview forum: "Scalable 3D Object-centric Learning"
_ICLR.cc/2023/Conference — Submitted to ICLR 2023_

### Official Review · Reviewer_bmFc · 2022-10-22

**Confidence:** 4
**Correctness:** 2
**Technical Novelty And Significance:** 3
**Empirical Novelty And Significance:** 2
**Recommendation:** 5

**Clarity, Quality, Novelty And Reproducibility:**

Some links to unsupervised learning from similar data representations and object localization are missing.

Unsupervised Generative Model-based 3D Object-centric Learning from 2D images:
[a] Toews, M., & Arbel, T. (2007, October). Detecting and localizing 3D object classes using viewpoint invariant reference frames. In 2007 IEEE 11th International Conference on Computer Vision (pp. 1-8). IEEE.
Specifically:
 - the viewpoint invariance mechanism is identical (i.e. a viewpoint-invariant reference frame defined in 2D and 3D by a location and orientation angle) to [a]
- the method of localizing object centers via Gaussian weighting is key to this method (Top of Page 5) log wg(xk) − SG(log wg(xk)),
      and is essentially similar to a difference-of-Gaussian used in [a]


**Details Of Ethics Concerns:**

None, limitations should be more accurately discussed.

**Strength And Weaknesses:**

Strengths
- The authors method is interesting and relevant, unsupervised learning of 3D objects in a scene is a challenging task.

Weaknesses
- The claims of novelty and unboundedness are overstated
- Some links to unsupervised learning from similar data representations and object localization are missing
- Experiments are limited to toy scenes, few objects


**Summary Of The Paper:**

The authors propose a scalable generative-model-based unsupervised 3D object-centric learning framework,
   that can learn scenes of arbitrary numbers of objects from multi-view camera observations.
A generative model is presented. Various experiments are performed on toy simulated dataset of a dozen basic 3D objects.

Unfortunately claims are overstated, e.g. scalability, similar classical methods are unreferenced.
 A large number of assumptions regarding the toy scene scenarios needs to be described in the limitations section.

**Summary Of The Review:**

The authors method is interesting, relevant and appears to work in the specific context, unsupervised learning.

Unfortunately the authors claims of novelty and model capabilities are missing references to similar prior work and ideas and overstated.
The claims of novelty are:
1) the first unbounded scalable generative-model-based unsupervised 3D object-centric learning framework.
2) view-invariant object representation
3) store a potentially unbounded number of objects detected for scalable inference

These are overstated considering the mathematical representation used and the assumptions of the method, as follows.

In my understanding, the model and mathematical data representation is as follows
In the 2D image plane:
   each object is described by a 2D coordinates location and orientation
     - images are assumed to be upright
In 3D:
    - objects are assumed to be upright, limited in number (not unbounded), lying on a plane in 3D
             thus described by 2D coordinates in a ground plane, and a single in-plane rotation.
   - in a fixed rigid configuration (e.g. no motion)

This representation is essentially mathematically equivalent to previous work in unsupervised viewpoint-invariant learning has been achieved with the same mathematical description[a], contradicting claims 1) and 2)

The authors then claim “We assume there are at most K objects”, which contradicts claims 1) and 3) of unboundedness.

* Limitations should be more accurately discussed. The authors state the only limitation is that objects out of view cannot be modeled.
    "We observe that our model may fail to model objects located on the boundaries of camera visual cones."
This is inaccurate, given the large number of limitations and assumptions, e.g. rigid scene, existence of a ground plane, upright camera, limited number of objects. The authors need to more accurately discuss the real limitations of the method.

* Related: the authors make use of the multiview NERF decoding representation.
A new method PERF, achieves similar accuracy to NERF with no training, via basic gradient methods.
[b] Rasmuson, Sverker, Erik Sintorn, and Ulf Assarsson. "PERF: Performant, Explicit Radiance Fields." arXiv preprint arXiv:2112.05598 (2021).  Frontiers in Computer Science 2022

---

> ### Author Response · Authors · 2022-11-17
> **Response to reviewer bmFC**
>
> - Claims are overstated
>   - We carefully read the paper pointed out by the reviewer. Based on our understanding, [a] is not deep learning papers and the goal is to match 2D portraits (one person in each image) taken from different angles.
>   - We'd like to make some clarification
> 	- We never claim to be the first work to use view-invariant formulation or the first one to learn view-invariant object representation and it is not a surprise that previous works have already exploited this idea. From the perspective of object-centric learning, we observe that the key to learning view-invariant object representation is to move the variational inference process into the camera coordinate system to remove any potential dependency on a global coordinate system.
> 	- The view-invariant object representation is fundamentally different from the reference paper [a] in the sense that our work learns a meaningful latent vector space (demonstrated in Fig 8). The representation learned by our model can be decoded into complete 3D geometries and further rendered into RGB, depth and instance mask for any views. In [a] the features are extracted from 2D images using rule-based algorithms (SIFT).
> 	- As stated in our paper, in our setting, we assume that **the total number of objects in each camera view** is bounded. However, **the total number of objects that exist in each scene** is unbounded, which is the first in object-centric learning literature. That is, we assume that our agent can observe at most K objects in each step. However, by navigating around the 3D scene, an unbounded number of objects can be sequentially discovered and stored for future queries.
> 	- Please note that our model receives RGBD images without any object instance information and our model learns to parse observation into objects and background. By contrast, there is no object-centric learning component or any generative component in [a] at all. Thus, our claim: "the first unbounded scalable generative-model-based unsupervised 3D object-centric learning framework" is valid and cannot be ignored because there exist works that store rule-based 2D feature points in a reference frame for future queries.
>
> -  log wg(xk) − SG(log wg(xk))
>    -  We failed to find related descriptions in [a]. It would be great if the reviewer can point out the exact content in [a].
> 	However, we'd like to point out that gaussian weighting is a common technique in object-centric literature [1, 2].
> 	The main novelty here is the gradient routing trick. To be more precise, the SG is the stop gradient operation.
> 	By + log wg(xk) we inject the gaussian prior into the gradient calculation but − SG(log wg(xk)) immediately to make sure the density value remains the same not to bias downstream computation and gradient computation. This gradient routing is inspired by the implementation of the Gumbel-softmax trick.
> 	Since [a] is not a deep learning paper, we find it hard to understand how log wg(xk) − SG(log wg(xk)) is not our innovation.
> 	Even log wg(xk) − SG(log wg(xk)) is not proposed by us first, we do not think that harms any contribution claimed in this paper.
>
> 	[1] Unsupervised Image Representation Learning with Deep Latent Particles
>
> 	[2] GENESIS-V2: Inferring Unordered Object Representations without Iterative Refinement
>
> - Missing reference
>   - We are more than happy to add the missing references.
>
> - Discussion of limitation
>   - No motion, rigid scene, existence of a ground plane, upright camera are all settings widely adopted by 3D object-centric learning literature [1, 2, 3, 4] and do not conflict with our claim.
>
>   [1] Decomposing 3D Scenes into Objects via Unsupervised Volume Segmentation
>
>   [2] Unsupervised Discovery of Object Radiance Fields
>
>   [3] Learning Object-Centric Representations of Multi-object Scenes from Multiple Views
>
>   [4] Spatially Invariant Unsupervised 3D Object-Centric Learning and Scene Decomposition
>
> - PERF method
>   - We thank the reviewer for pointing out the interesting PERF method. We'd like to point out that PERF is at  the current stage a per-scene data representation while our work learns a generalizable NeRF (one inference pipeline for multiple objects). But we certainly think PERF could be a strong candidate for geometric representation for future object-centric learning methods.

---

> > ### Comment · Reviewer_bmFc · 2022-11-27
> > **Response**
> >
> > Thanks to the authors for the response. I think this work is interesting and relevant. However a substantial revision is necessary to adjust claims of novelty, missing related literature using identical mathematical representations ( 'deep learning ' or other technologies), and also practical application on real images.

---

> > > ### Author Response · Authors · 2022-12-06
> > > **Response**
> > >
> > > Thanks for your feedback.
> > >
> > > After carefully reading the paper [a] again, we have to disagree with the assertion that [a] and our work uses "identical mathematical representation". As stated in the previous response, our work, built on an **iterative amortized variational inference** framework, tackles the unbounded object-centric learning problem, which is drastically different from the problem and the approach described in [a].

---

### Official Review · Reviewer_a3V1 · 2022-10-24

**Confidence:** 4
**Correctness:** 2
**Technical Novelty And Significance:** 3
**Empirical Novelty And Significance:** 3
**Recommendation:** 3

**Clarity, Quality, Novelty And Reproducibility:**

The paper is generally well written and clear. The method builds on solid existing frameworks and methods (i.e. VAEs, Obsurf for its compositional nature and training-with-depth procedure and NeRF). I think this combination can be important to obtain practical and useful models. The framing of the method as a generative model, while having varying number of latents that aren't handled by the model, limit its technical soundness. It would be useful in order to understand the model to provide a diagram of the generative model and inference as is typically done in the literature.
The proposal of using the Cognitive Map is novel and the experimental results are significant. The pseudo algorithms provided and training details make this work reproducible.

**Strength And Weaknesses:**

**Strengths**
- Literature in unsupervised learning commonly focus on scenes that fit within the camera's field-of-view which is a strong limitation. This work tackles scenes in which the stream of input views do not have that property.
- Latent representation is disentangled and interpretable to a large extent due to its $z_{where}, z_{what}, z_{pres}$ structure. This enables better control the scene useful for downstream tasks.
- Cognitive map is a novel mechanism to scale the model to an unbounded number of objects.
- Experiments clearly show the ability of the model to handle larger scenes at test time.

**Weaknesses**
1. While the authors refer to their method as a generative model, proposing a model with variable sets of latents has some technical requirements, otherwise it is not well defined. In this case, I do not see how the proposed method can sample scenes and images because there is no explicit distribution over e.g. the number of objects per scene.  Since the model cannot control the number of objects of a scene, this is instead dealt in an ad-hoc way through its inference mechanism. For instance by choosing $q(z_{pres} = 1) < 0.5$ to be the threshold of discarding presence.
2. Related to the point above, the authors should be clearly explain how the discrete distribution over $z_{pres}$ (Bernoulli) is learned. Due to its non-differentiable nature, what is the signal to learn the logits? Do they use continuous relaxation, REINFORCE, or some other approximation (or rather sum the latents out)?
3. SOOC3D requires a first pre-training stage (curriculum learning) to work well in practice and should be referenced from the description of the method in the main text.
4. A particular weakness to the method, acknowledged by the authors, is that partially viewed objects are not handled properly when querying the cognitive map. This is an issue that specially arises as a consequence of the choice of query mechanism by object position and it seems to be a potentially important limitation in many types of scenes.
5. This work assumes RGBD data and requires precise camera information (due to use of NeRF), this may limit its applicability to only some datasets. An ablation of the method without using depths (as the model doesn't technically seem to require depths) would be useful.
6. The refinement network takes decoded NeRF images as part of its input, this can result in expensive inference as rendering with NeRF can be notoriously slow. Can the authors comment on the computational efficiency of inference at test time?
7. As this work attempts to tackle larger-scale scenes, it seems important to experiment on more realistic datasets that have (a) larger and cluttered number objects, and (b) more complex backgrounds (which is often a challenge with self-supervsied segmentation methods). An example of such is the MultiShapenet (MSN) dataset used in [1].

**Further questions/suggestions**
1. What type of rendering is used with NeRF at test time in the experiments? Does the NeRF fine-tuning process use the train-with-depth loss or is it based on volumetric rendering loss? (this question is for both training and test time)
2. Have the authors experimented with an ablation without the Cognitive Map? Does it result in worse/better performance?
3. While the authors suggest that the model performs on par with Obsurf on CLEVR3D and the
MultiShapeNet, the numbers seems to show a consistently worse performance. Would the authors elaborate what might explain the difference in performance?
4. The deterministic slot attention baseline is great but it would be great to provide further reconstruction results to establish the usefulness of the method's iterative variational inference. While they state identities are not well preserved, this may not be an issue if the segmentations at the final iteration are strong (FG-ARI), please provide those results if available.

[1] Sajjadi et al. Scene Representation Transformer: Geometry-Free Novel View Synthesis Through Set-Latent Scene Representations. CVPR 2022


**Summary Of The Paper:**

This work presents a method for unsupervised object-centric learning from multiple images. The method is based on a number of techniques: generative modelling with variational inference and neural radiance fields (NeRF) to obtain 3D scene representations and rendering. In contrast to existing compositional and amortized NeRF models, this work proposes a latent object representation, called Cognitive Map with registration and querying mechanisms that allow inferring and reconstructing an unbounded number of objects. Results show how the method capable of segmenting the objects for larger scenes (in terms of spatial extent and number of objects) than those seen at training time, while other methods in the literature fail to generalize in such a way.

**Summary Of The Review:**

SOOC3D allows obtaining a highly interpretable and controllable 3D object-centric representations from images. In contrast to literature, it tackles scenes that have variable (and possibly unlimited) number of objects using its novel Cognitive Map. Instead, most object-centric methods assume a fixed number of maximum number of objects. These are key strengths of this work that to my knowledge no other methods have. However, it has some theoretical and experimental limitations: The method posed as a generative model doesn't seem to be a proper model of the scenes it tackles. It's also unclear how the distribution over the binary presence latents is learned, its probabilities are used in an arbitrary way, and do not handle partially-viewed objects.
The method also needs a pre-training stage to make it work which makes it less useful for the practitioner.  Finally, as the literature is shifting towards more complex and real images, it would be useful to test the limits of this model on such data regimes (for instance on datasets such as the MSN dataset mentioned above). Unless the authors address my questions regarding its soundness, I believe this paper needs to address these design issues before being accepted.

---

**Post-rebuttal update**

I thank the authors for addressing my questions in detail and the changes to the paper they commit to doing. I think they authors have a novel and interesting paper, and propose a number of useful techniques to make their method scalable. I think the contributions have the potential to being useful to the community.
However, it still needs substantial changes before being ready.

Discrete latent variables of object presence are introduced as part of the iterative variational inference model, but the issues that this entails is not fully addressed. The authors refer to using a relaxed bernoulli distribution (which is still not cited nor explained in the updated manuscript) but little to no details are given (i.e. no annealing of the relaxed Bernoulli termperature, etc., use of non-differentiable terms, etc.). Instead, the authors claim that the distribution learns accurate presence probabilities thanks to the learning signal from the KL and log-likelihood terms. These claim does not seem substantiated. The presence distribution seem to be mostly learned during a pre-training phase using a hand-crafted simpler version of the datasets.
This work's main contributions would therefore be stronger if either:
- the authors fully address the challenges of discrete modelling of objects (cf learning signal) and show the method works without pre-training.
- and/or the method can be shown to work with substantially more challenging datasets. I do not think that dataset appearance complexity is orthogonal to the method in this work, since such increased complexity (textures, clutter, occlusion) complicates learning disentangled object representations (what, where, presence).

SOOC3D+ relies on fitting a NeRF scene post-generation, and uses depth supervision loss that is also during training. As far as I understand, results and renders shown for SOOC3D+ therefore leverage privileged information (depth) to fit the scenes, which is unfair when comparing to methods that do not use depths.
Specifically, SOOC3D has worse RMSE than MulMon without finetuning, contradicting the claim that it complete outperforms MulMon.
Regular volumetric rendering likelihood should be used in this finetuning stage as we cannot generally assume ground-truth depths at test time.

---

> ### Author Response · Authors · 2022-11-17
> **Response to reviewer a3V1**
>
> - Not a generative model
>   - With due respect, we disagree with the reviewer and firmly believe that our model is indeed a generative model. For the scalability purpose, we made an independent assumption between objects. Please note that such an assumption is common in generative object-centric learning literature for images [1, 2, 3] or point cloud [4]. More works can be listed if required. Particularly, both [1] and [4] infer a z_pres for each object candidate.
> 	We'd like to point it out that all works mentioned above as well as ours fail to model the joint distribution over z_pres.
>     Please note that, while we assume z_pres to be independent from each other, we can run our trained model on the training set and easily estimate the distribution via a frequency estimator. Such disentangled prior estimation can be seen in previous works [5].
>
> 	More importantly, our work aims at parsing scenes with unbounded scales. Modeling a distribution over a potentially unbounded number of objects is not practical.
> 	Since z_pres is a binary variable, q(z_pres = 1) < 0.5 indicates that z_pres is more likely to be 0. Thus, thresholding at 0.5 is not an ad-hoc inference, but follows the maximum likelihood principle.
>
> 	[1] SPACE: Unsupervised Object-oriented Scene Representation via Spatial Attention and Decomposition
>
> 	[2] Multi-Object Representation Learning with Iterative Variational Inference
>
> 	[3] Learning Object-Centric Representations of Multi-object Scenes from Multiple Views
>
> 	[4] Spatially Invariant Unsupervised 3D Object-Centric Learning and Scene Decomposition
>
>      [5] Neural Discrete Representation Learning
>
> - How z_pres is learned
>   - During training, we use continuous relaxation to learn this latent.
>
> - Curriculum learning
>   - We will modify our paper and highlight this in the main text.
>   - While a curriculum learning stage is generally regarded as a nontrivial process during training, we'd like to point out that our curriculum learning step does not require any modification to our inference pipeline. Therefore, it is still considered as a reasonable design for our work as the first attempt in achieving unbounded scalability in inference.
>
>
> - Partially viewed objects
>   - We mentioned in the main paper that this limitation can be fundamentally resolved by predicting bounding boxes.
> 	However, bounding box prediction requires a re-structure of our latent space, which is non-trivial. Thus, we leave it as our future work. At the current stage, the limitation can be easily alleviated by enlarging the cognitive map query FOV (field of view) or adding a margin to the query region.
> 	To give a concrete example, when working with a camera of FOV = 90 degrees, we can query the cognitive map with FOV = 120 degrees.
> 	It also makes sense to retrieve objects that are not observed completely since there will be no gradient flow coming from the observation and the KL term encourages all latents to remain the same.
> 	While more complicated query criteria can be designed and applied, it will not change our inference process or affect the training process in any way.

---

> > ### Author Response · Authors · 2022-11-17
> > **Response continue**
> >
> >
> > - RGBD data and NeRF rendering time complexity
> >   It is known that vanilla NeRF rendering is both time-consuming and memory-consuming. Considering the iterative nature of our model, vanilla NeRF is simply infeasible during both training and inferences. As detailed in Appendix B, RGBD information allows fast likelihood computation. In short, under the mild assumption that each ray will scatter (hit something) eventually, the NeRF rendering equation induces a probability distribution over the hit distance. With depth information, we can directly compute the likelihood of the hit point of each ray and maximize the likelihood w.r.t NeRF parameters.
> >   We'd like to highlight the fact that, even with RGBD data, our work is the first to handle unbounded scenes.
> >   Lifting the requirement on depth as input is an interesting and meaningful future research direction. However, we believe it is out of the scope of this work.
> >
> > - The dataset is simple.
> >   - We agree that the dataset is visually simple. However, our main focus is scalability, which is orthogonal to visual complexity. From a scale perspective, the dataset is really challenging and no previous methods can tackle the problem on such large scale data.
> >   - In the related work section, we discussed the gap between ours and previous works in detail.
> >   - We also demonstrate our model on the blender dataset to show that the scalability gain does not sacrifice object modeling quality on visually non-trivial scenes.
> >   - Note that our model already outperforms MulMON, which is the previous state-of-the-art.
> >
> > - Rendering
> >   - During training, we simply estimate the ray likelihood as described in Appendix B.
> >   - During testing, we render novel views with vanilla NeRF rendering equations.
> >   - During fine-tuning, we optimize the same ray likelihood function as that during training.
> >
> > - Ablation without cognitive map
> >   - Cognitive map stores all previously discovered object latents and for each input or query view only retrieve objects in camera cones.
> >   - For small scale scenes, removing cognitive maps falls back to the curriculum learning setup and the performance is similar.
> >   - For medium and large scale scenes, removing the cognitive map means that all previously discovered object latents are sent into the inference process and the memory consumption grows larger than what the computing resources can handle.
> >   - We conjecture that the performance will be slightly better since objects on the cone boundaries are also retrieved without the constraint on the memory consumption.
> >
> > - Performance compared with ObSuRF
> >   - First, please note that as stated in the appendix, the dataset that ObSuRF is tested on only supports one input view and one query view setup and there are not enough views for multi-view update which is required by our method. However, we test our model to one input view and one query view setup for fair comparison. Note that our model is not intended for such a setup.
> >   - We can notice that the Fg ARI is better but the ARI is slightly worse. This is due to the explicit modeling of position and rotation of each object. When there are uncertainties about object poses, our model tends to enlarge the volume of each object slightly to compensate for the uncertainties.
> >
> > - Slot attention usefulness
> >   - Please note that our work focuses on end-to-end scalable inference. If identities are not preserved, one needs to manually match objects before and after each update.
> > 	Compared with the requirement for a curriculum learning stage, we think object matching across images is much more troublesome.
> > 	One contribution of our work is to remove the requirement for any ad-hoc, heuristic-based manual object identity matching and let the aromatized variational inference to handle the object matching automatically.

---

> > > ### Comment · Reviewer_a3V1 · 2022-11-18
> > > **Response to authors**
> > >
> > > I appreciate the author's detailed answers that have addressed a number of my questions or concerns.
> > > Some points are still unclear, however:
> > >
> > > **Generative model**
> > > I agree that the model is for the most part well defined and follows independence assumptions in line with other works. My issue was not with the independence assumption, but with the claim that the generative model models scenes with unbounded number of objects.
> > > One way to understand a generative model is to ancestrally generate samples from the prior and check that the sampled observations are similar to that of the data. When I apply this to the proposed model, I'm not sure we get observations of scenes with varying number of objects. That is, sample $z_0 \sim p(Z_0)$, sample $x_0 \sim p(X_0|z_0)$, sample $z_1 \sim p(Z_1|x_0)$, sample $p(X_1|z_1)$, etc. Unless real observations are given to the model, I don't see how the generated sequence $x_0$, $x_1$, etc. will contain newly instantiated objects. Note that the model has a prior $p(Z_{pres} = 1) \approx 0$, and only objects are generated in $x_t$ when there are latent objects present.
> > > If the model cannot be unrolled in such a closed loop way, and instead needs to condition its latent model with real observations (e.g. to have new objects appear), then it doesn't seem to be fully specified as a generative model.
> > > This is not necessarily a bad thing if the model still serves its purpose (scene inference), but the authors should make these things clear in the paper. For instance, a reader might think this model can be unrolled to generate scenes and use for e.g. planning which wouldn't be possible if it is not defined properly.
> > >
> > > **Discrete presence latent**
> > > > During training, we use continuous relaxation to learn this latent.
> > >
> > > Can the authors elaborate on this? What type of continuous relaxation is used and what are the details? For example, is there a temperature annealing through training to avoid biased gradients when combining it with reparameterization? Is this applied in all stages of the curriculum?
> > > In addition to how sampling $z_{pres}$ is handled, one of the key designs of the model is the registration of objects in the cognitive map via $q(Z_{pres} = 1|...) < 0.5$. Again this is a non-differentiable function, which means the posterior $q(Z_{pres}|...)$ does not get learning signal in this way. Can the authors explain where does the signal to learn $q(z_{pres})$ come in?
> > > At the moment, it seems that this posterior is only learned in the first stage of the curriculum, which requires a simplified dataset to do so. If this is the case, this is a strong limitation and the work would improve if the proposed method allowed learning the presence distribution using the full datasets.
> > >
> > > **Registration**
> > >
> > > Speaking of registration, why is presence decided based on $q(Z_{pres} = 1|...) < 0.5$ and not by sampling from $Q(Z_{pres}|...)$?
> > > As it is, there is no gradient flowing through the decision to register.
> > > Also, where is the sampled $Z_{pres}$ used if at all. Is it used in the refinement network?

---

> > > > ### Author Response · Authors · 2022-11-18
> > > > **Response to reviewer a3V1**
> > > >
> > > > - Generative Model
> > > >   - Due to the interpretation of z being the existence of one object, we do not need the ancestral structure p(x | z) since when z = 0, the entire x will be turned off. Thus, we can sample x and z independently and delete x whenever the corresponding z = 0.
> > > >   - Due to the independence assumption between objects (which is the key to extreme scalability) we do not model the joint distributions over z_pres and z_where. While one can manually learn a joint distribution on z_pres and z_where using the inference results on the training set for sampling. we agree that it is not straightforward to "unroll" our model. We will clarify this point in our paper.
> > > >
> > > > - Discrete Presence Latent
> > > >   - In our implementation, we use RelaxedBernoulli from PyTorch with temperature = 1 with no annealing. We observe that our training objective will lead to a really sharp q(z_pres | ...). We agree that with annealing, the training process might be more stable.
> > > >   - Note that, after discarding objects with q(z_pres | ...) < 0.5, for the remaining objects the proposal distribution q(z_pres | ...) is also registered into the cognitive map to receive gradient flow. The learning signal comes from both KL and likelihood. (1) The KL divergence term encourages z_pres to be close to zero (we refer the reviewer to Appendix B for detailed discussion). (2) The likelihood term encourages z_pres to be close to one (we refer the reviewer to Sec 3.2 NeRF decoding for detailed discussion).
> > > >   - During the first stage of curriculum learning, we limit the training data to small scenes. We do not use the cognitive map (thus limiting the total number of objects in scenes to K) and no objects will be rejected/deleted (but can still be ignored/negated with z_pres = 0). After the first stage of curriculum learning, q(z_pres | ...) will be sharp and accurate and serve as a good initialization for the next training stage.
> > > >   - Yes, the z_pres learning process is not aware of the cognitive map rejecting process. However, as a result of the first stage of curriculum learning, the z_pres proposal distribution correctly reflects the existence of objects and assigns q(z_pres = 1| ...) ~= 0 to objects that should not exist. Thus, discarding objects with z_pres = 0 does not introduce a large bias into the learning process since they are practically discarded already. As stated above, the KL term and likelihood term encourage the z_pres proposal distribution to accurately reflect the existence of objects. Thus, our model still learns well even being agnostic to the cognitive map rejection.
> > > >   - Please note that, as a special property of the intended task, the "simplified" dataset is only a sub-scene of a full scene. That is, unlike other tasks where a set of "simpler" tasks need to be built specially, no separate data collection phase is needed for our task. Thus, we do not think the limitation is strong.
> > > >   - We indeed share the same wish of the reviewer, i.e. making the pipeline as simple as possible. However, as a first step on the scalability problem, we believe that it is fair to leave the improvement pointed out by the reviewer for future works.
> > > > - Registration
> > > >   - It is still possible to sample z_pres = 0 when q(z_pres = 1) >> 0.5. If an object is rejected, all the information is lost and needs to be rediscovered. Thus, to reduce the variance, the registration is based on q(z_pres = 1) instead of z_pres. The sampled z_pres is used in the likelihood computation (Sec 3.2 NeRF decoding the 3rd line page 5.).
> > > >   - Note that z_pres is sample and q(z_pres) is expectation. In some cases, they can be used interchangeably and which quantity to use are specific design choices and are both correct.

---

### Official Review · Reviewer_1tCi · 2022-10-24

**Confidence:** 4
**Correctness:** 3
**Technical Novelty And Significance:** 3
**Empirical Novelty And Significance:** 3
**Recommendation:** 5

**Clarity, Quality, Novelty And Reproducibility:**

From my viewpoint, the paper is of high novelty. But it lacks clarity and thus decreases reproducibility.

**Strength And Weaknesses:**

## Strength
- Scalable object-centric learning is very interesting and novel to the community.
- The experiments in the synthetic dataset are promising.

## Weaknesses.
The paper has several weaknesses which I'll detail below:
### Writing
Maybe I missed something, I have difficulties following the method details. Specifically,
- I'm not sure about the role of amortized variational inference. If the method models the object individually, why is this step needed?
- In sec 3.1, the paper claims the presented method is a generative model. But this is confusing to me. Because I didn't see any description of latent space. Is this an auto-decoder/generative latent optimization since I didn't see any encoder?
- The workflow is not very clear to me.
I would recommend restructuring the method section. Particularly, I would recommend better motivation when proposing a new module. This might improve the clarity a lot.

### Method
The paper claims the advantage of dealing with large-scale scenes compared with other works. However, this work has lots of limitations. It's limited to simple objects, unlike other works that model realistic datasets.

### Experiments
- The dataset shown in the paper is too simple. This makes doubt the ability of the presented method. I would recommend having more photorealistic datasets like Kubric. It would be convincing.
- Given that the proposed method is a generative model, I would be curious about the latent space. Interpolation analysis might be helpful here.



**Summary Of The Paper:**

The paper presents a framework for scalable 3D object-centric learning. Unlike existing works that are limited to the bounded scene, this work allows to model 3D objects present in the large-scale 3D scenes. Specifically, the method mains a cognitive Map that allows the registration and querying of objects on a global map. This enables modeling 3D models in large-scale scenes. For each 2D observation, the method learns to radiance method in an object-centric coordinate system. Furthermore, the method models the object individually. The paper shows promising results on a synthetic dataset.

**Summary Of The Review:**

The presented method has a good motivation -- aiming at scalable 3D object-centric learning. But however, the paper is of low clarity. And more importantly, the experimental results/analysis do look not very strong as I detailed above. I thus tend to reject this paper. But I would like to hear back from the author during the rebuttal just in case I misunderstand anything.

---

> ### Author Response · Authors · 2022-11-17
> **Response to reviewer 1tCi**
>
> - Why amortized variational inference is used
>   - To deal with sequential input, we adopt amortized variational inference framework.
> 	To be more specific, a traditional VAE takes the form of an encoder-posterior-decoder.
> 	The amortized variational inference framework takes the form of prior-decoder-reconstruction-refinement-latent_delta-posterior recurrently (posterior is the prior for the next iteration).
> 	The initial latent is sampled from a global prior and is decoder to reconstruct observations.
> 	Then a refinement network takes the reconstructed observations, reconstruction loss, etc to update prior into posterior.
>   - Compared with a vanilla VAE, the amortized variational inference is better suited for sequential data due to its iterative nature.
>   - The amortized variational inference is used to factorize RGBD observations into a set of object latent variables. Without amortized variational inference processes, we cannot model objects individually since there will be no concept of "object".
> - description of latent space
>   - The description of latent space can be found in section 3.1 including the latent space dimension, interpretation as well as the distribution parameterization.
>   - Section 4.2 shows the quality of our learned representation by latent space traversal.
> - Where is the encoder
>
>   - See Appendix A for an algorithmic summary.
>
> - The dataset is simple.
>   - We agree that the dataset is visually simple. However, our main focus is the scalability, which is orthogonal to visual complexity. From the scale perspective, the dataset is really challenging and no previous methods can tackle the problem.
>   - In the related work section, we detailed the gap between ours and previous works carefully.
>   - We also demonstrate the effectiveness of our model on the blender dataset to show that the benefit of the scalability modeling does not sacrifice object modeling quality on visually non-trivial scenes.
>   - Our model has already outperformed MulMON, which is the state-of-the-art method.

---

### Official Review · Reviewer_zLxX · 2022-10-26

**Confidence:** 3
**Correctness:** 3
**Technical Novelty And Significance:** 2
**Empirical Novelty And Significance:** 2
**Recommendation:** 5

**Clarity, Quality, Novelty And Reproducibility:**

The paper is mostly well-written and the formulation appears to be correct.
The novelty is questionable given the similarity to the prior work MulMON.
The lack of details related to 3D geometry would make it hard to reproduce the results of this paper.

**Strength And Weaknesses:**

Incrementally learning object representations over time in a multi-object setting is an important and timely problem. This paper does a good job in formulating the problem as a latent probabilistic inference. The paper describes several non-trivial concepts quite lucidly.

The main issue I noticed was the similarity to the prior work MulMON. I read both papers carefully, and I find only incremental modifications in the paper.
The ideas presented as novel aspects in the paper are also not clearly motivated/explained. Particularly, the camera pose and the NeRF models are expressed through latents, but how are they connected to the underlying 3D geometry? The formulation appears to be high-level, and might not be relevant to real-world problems. It would be useful to make connections to practical techniques such as iMAP (ICCV 2021), which also look to solve the incremental scene mapping problem.
The paper also claims to be able to handle unbounded scenes. But there are assumptions that appear to violate that flexibility. For example, the method uses RGBD data, (instead of RGB that is typically used in NeRFs) and truncates depth values larger than the clipping plane. It also uses global camera pose, which implicitly means that the scene is bounded.



**Summary Of The Paper:**

The paper proposes a 3D object-centric representation learning on potentially unbounded scenes. The method infers object poses and view-invariant object representations in object coordinate system using RGBD data. Amortized variational inference is used to process sequential input and update object representations online, and formulated as an ELBO objective. The results are presented on synthetic datasets and the metrics show that the method outperforms MulMON, a prior art for 3D object centric learning.

**Summary Of The Review:**

The paper attempts to solve a important problem of incremental object learning in multi-object setting. However, the contribution to the learning method is incremental. given prior work. The probability densities should be connected to the underlying object and scene geometry and the formulation should be explained clearly. Finally, more ablation studies could be used to explain individual design choices.

---

> ### Author Response · Authors · 2022-11-17
> **Response to reviewer zLxX**
>
> - Main contribution and motivation
>   - The iterative amortized refinement model has been studied extensively in the literature [1, 2, 3] and is not claimed as our contribution.
>   - None of our contributions claimed in Section 1 overlap with that of MulMON. Therefore, we believe our proposed method has sufficient novelty.
>
> [1] Iterative Amortized Inference
>
> [2] A General Method for Amortizing Variational Filtering
>
> [3] Multi-Object Representation Learning with Iterative Variational Inference
>
> - Connection to iMAP
>   - iMAP takes RGB-D data as input and reconstructs 3D scene geometry.
>   - The major difference is that our model focuses on object-centric scene understanding and the inference of object instance information which is not tackled by iMAP.
>   - In particular, we focus on Inferring object instance segmentation in an unsupervised manner which itself is a challenging task.
>   - While iMAP incrementally refines scene geometry, our model incrementally **discovers and refines** object’s latent representations.
>
> - How are they connected to the underlying 3D geometry?
>   - Our decoder network decodes each object’s latents into a NeRF, which is supervised by ground truth RGBD observation.
>   - That is, each object latent variable induces a probability distribution over the object geometries via our decoder.
>   - Our depth reconstruction results demonstrate that our objects’ latents encode accurate geometrical information as well as appearance information.
>   - Our inferred object and scene embeddings can be directly used as input for downstream tasks like relational reasoning.
>   -Our method can output 3D scene geometry by decoding our object and scene latents as NeRF, which can be further converted into other geometry representation like mesh or point cloud easily.
>
> - Assumptions and unbounded scenes
>   - RGBD data
> 	- NeRF-based methods that support novel view synthesis from an arbitrary view normally require a large number of RGB input views to accurately infer the object geometries. In contrast, with the depth information, our model only requires a small number of views for training.
> 	- Also, it is well known that vanilla NeRF rendering is both time-consuming and memory-consuming. Considering the iterative nature of our model, vanilla NeRF is simply infeasible during training and inference time. As detailed in Appendix B, RGBD information allows fast likelihood computation.
>   - Truncated depth value
> 	- We'd like to point out that while the camera cone of each view is bounded, the scale of scenes that our model can process is unbounded, which is the first approach proposed in the object-centric learning literature. The main contribution of our work is the ability to integrate information from a diverse set of views (with bounded camera cones) to form a complete object-centric representation of a scene of arbitrary scale. Also note that for novel view rendering, one can lift the clipping plane constraint and render each ray towards an unbounded depth (with huge computational cost of course).
>   - Global camera pose
> 	-  Our model works with **an arbitrary global coordinate system** not **a specific one**. In other words, our model only requires the relative poses between cameras. One can easily set the origin of the global coordinate system to be at the first input camera view or even randomly, at the beginning of one inference.
> 	- The global coordinate system is never the direct input to our model but only used for object registration and query. Thus, our model will not develop any dependency on the value of the exact coordinate.
> 	- Thus, the global camera pose assumption does not contradict with the unbounded scalability.
>
> - Ablation study
>   -  In the paper, we present the most simplified structure of our pipeline. It means every component in the model is necessary to successfully detect and reconstruction objects. Thus, no ablation study is provided in the paper.

---

### Decision · Program_Chairs · 2023-01-20

**Decision:**

Reject

**Justification For Why Not Higher Score:**

Needs major revisions in terms of clarity, rigorous formulation of all design choices and more realistic domains for experimental validation.

**Justification For Why Not Lower Score:**

N/A

**Metareview: Summary, Strengths And Weaknesses:**

Object-centric learning is typically done in a 2d/latent space to learn disentangled representations from sensory observations. This paper studies the problem within a 3D context with a major relaxation compared to previous work -- that there could be an unbounded number of objects in the scene during test time (but assumes upto K objects in current view). This is the problem of 'object permanance', a long standing open problem in computer vision and cog sci/neuroscience. There is currently no known solution to this problem, even in toy domains. So at a high level, I think think this problem is quite interesting for the community to tackle with a lot of real world implications.

This model proposes an approximate generative model using variational inference and neural radiance fields. In order to handle object permanance over a potentially unbounded set of objects, authors propose a 'Cognitive map' inspired architecture to register and track representations over viewpoints. The end result is that the method is able to segment multiple objects in a scene, with the corresponding radiance field for each object. All experimental results are show on toy rendered scenes.

There is consensus that the proposed approach studies a novel setting unlike prior approaches and shows reasonable experimental validation to provide its effectiveness to some extent.

However, many questions remain unanswered before the paper is ready for publication. For instance there were questions about the generative nature of this approach, rigorous foundations for why certain choices were made and clarity of writing. While it is informative and necessary to study a toy domain, I would encourage the authors to perform experiments in some real image dataset as suggested by the reviewers. Since the approach is pitched to work with unbounded scenes, I think it is necessary to show some scaling on real world domains, even if they were directional. Otherwise it will be difficult for anyone, including the authors, to have conviction in all design choices and also make it difficult for other researchers to build on this paper.

Overall I believe the paper studies an extremely important and interesting problem but it is not yet ready for publication.